# Endophilin A1 facilitates organization of the GABAergic postsynaptic machinery to maintain excitation-inhibition balance

**Xue Chen[1,2], Deng Pan[1,2], Jia-Jia Liu[1,2]\*, Yanrui Yang[1]\***

[1]State Key Laboratory of Molecular Developmental Biology, Institute of Genetics and Developmental Biology, Chinese Academy of Sciences, Beijing, China; [2]College of Life Sciences, University of Chinese Academy of Sciences, Beijing, China

## eLife Assessment

This study presents a **valuable** finding on the molecular mechanisms that govern GABAergic inhibitory synapse function. The authors propose that Endophilin A1 serves as a novel regulator of GABAergic synapses by acting as a component of the inhibitory postsynaptic density. The findings are **convincing** and likely to interest a broad audience of scientists focusing on inhibitory synaptic transmission, the excitation-inhibition balance, and its disruption in disorders such as epilepsy.

**\*For correspondence:**
jjliu@genetics.ac.cn (J-JL);
yryang@genetics.ac.cn (YY)

**Competing interest:** The authors declare that no competing interests exist.

## Abstract

The assembly and operation of neural circuits in the brain rely on the coordination and balance of excitatory and inhibitory activities. Inhibitory synapses are key regulators of the functional balance of neural circuits. However, due to the diversity of inhibitory presynaptic neurons, the complex composition of postsynaptic receptor subunits, and the lack of typical postsynaptic dense structure, there are relatively few studies on the regulatory mechanisms for inhibitory synaptic structure and function, and insufficient understanding of the cellular and molecular abnormalities of inhibitory synapses in neurological and neuropsychiatric disorders. Here, we report a crucial role for endophilin A1 in inhibitory synapses. We show that endophilin A1 directly interacts with the inhibitory postsynaptic scaffold protein gephyrin in excitatory neurons and promotes organization of the inhibitory postsynaptic density and synaptic recruitment/stabilization of the γ-aminobutyric acid type A receptors via its plasma membrane association and actin polymerization-promoting activities. Loss of endophilin A1 by gene knockout in mouse hippocampal CA1 pyramidal cells weakens inhibitory synaptic transmission and causes imbalance in the excitatory/inhibitory function of neural circuits, leading to increased susceptibility to epilepsy. Our findings identify endophilin A1 as an iPSD component and provide new insights into the organization and stabilization of inhibitory postsynapses to maintain E/I balance as well as the pathogenesis of epilepsy.

## Introduction

The excitatory and inhibitory synaptic activities of neural circuits are tightly orchestrated in the central nervous system (*Sakimoto et al., 2021*). The imbalance between excitatory and inhibitory neuronal activity is implicated in many neurological and neuropsychiatric diseases including schizophrenia, autism spectrum disorder, and epilepsy (*Canitano and Pallagrosi, 2017*; *Canitano and Palumbi, 2021*; *Foss-Feig et al., 2017*; *Fritschy, 2008*). In the brain, in contrast to asymmetric excitatory synapses formed between axonal boutons and dendritic postsynaptic structures with thick disc-shaped postsynaptic density (PSD), inhibitory synapses are symmetric and are constituted of the presynaptic terminal from diverse interneurons and the postsynaptic plasma membrane with thin sheet-like PSD where receptor

complexes are anchored (*Bai et al., 2021*; *Favuzzi et al., 2019*; *Liu et al., 2020*; *Mody and Pearce, 2004*; *Tao et al., 2018*). Although the formation, maturation, and plasticity of both excitatory and inhibitory synapses are vitally important for neural circuit functioning, much more research effort has been focused on the regulatory mechanisms governing excitatory synaptic structure and function. The mechanisms by which inhibitory synapses are constructed in the brain and fine-tuned in response to neural activity remain largely unexplored.

The γ-aminobutyric acid type A receptors (GABA$_A$Rs), major targets of antiseizure drugs, are hetero-pentameric chloride channels that mediate fast synaptic transmission at inhibitory synapses in the brain (*Scott and Aricescu, 2019*). Several neuropathologies are characterized by alterations in GABA$_A$ receptor trafficking, synaptic E/I ratio, and neuronal excitability (*Sakimoto et al., 2021*). At the inhibitory postsynaptic site, synaptic localization and functioning of GABA$_A$Rs requires trans-synaptic cell adhesion molecules, submembranous scaffold proteins, and intracellular signaling proteins. Gephyrin is a master organizer of iPSD, which self-assembles into planar lattices beneath the plasma membrane and anchors GABA$_A$Rs or glycine receptors (GlyRs) to the postsynaptic sites (*George et al., 2022*; *Kasaragod and Schindelin, 2019*; *Kirsch et al., 1991*; *Tyagarajan and Fritschy, 2014*). Collybistin, a membrane lipid-binding protein, functions in recruitment of gephyrin and GABA$_A$Rs to the plasma membrane by interacting with gephyrin, the synaptic adhesion molecule neuroligin-2 (NL2) and the α2 subunit of GABA$_A$Rs (*Kalscheuer et al., 2009*; *Kins et al., 2000*; *Poulopoulos et al., 2009*; *Saiepour et al., 2010*). The transmembrane protein Shisa7 binds GABA$_A$Rs and regulates their trafficking, synaptic abundance, and functional properties via binding to protein(s) other than gephyrin (*Han et al., 2019*; *Liu et al., 2020*). Nevertheless, mechanisms governing the assembly, recruitment, and maintenance of various inhibitory synaptic components remain largely enigmatic.

The majority of inhibitory synapses are formed on the dendritic shaft, or around the cell body of excitatory principal neurons where both actin and microtubule cytoskeletons are present underneath the plasma membrane. Through direct interaction with polymerized tubulin, gephyrin organizes the iPSD by anchoring the GlyRs and GABA$_A$Rs to the microtubule cytoskeleton (*Parato and Bartolini, 2021*). Notably, although small F-actin patches are observed within the dendritic shaft where the majority of inhibitory synapses are assembled (*van Bommel et al., 2019*), role(s) of the actin cytoskeleton in the organization and maintenance of inhibitory synapses is still unclear.

Endophilin A1 (EndoA1) is a member of the endophilin A protein family characterized by an amino (N)-terminal membrane-binding N-BIN/amphiphysin/Rvs (BAR) domain and a carboxyl (C)-terminal Src homology 3 (SH3) domain. The gene encoding endophilin A1 (gene name *sh3gl2*) is almost exclusively expressed in the brain (*Ringstad et al., 1997*), and has been implicated in epilepsy, schizophrenia, depression, Parkinson's disease, and Alzheimer's disease (*Chang et al., 2017*; *Corponi et al., 2019*; *Germer et al., 2019*; *Ren et al., 2008*; *Yu et al., 2018a*; *Yu et al., 2018b*). In excitatory synapses, endophilin A1 functions in synaptic vesicle recycling and autophagosome formation at the axon terminal (*Bademosi et al., 2023*; *Milosevic et al., 2011*; *Murdoch et al., 2016*; *Ringstad et al., 1999*; *Schuske et al., 2003*; *Soukup et al., 2016*; *Verstreken et al., 2003*; *Watanabe et al., 2018*), and regulates the morphogenesis, maturation, and acute structural plasticity of dendritic spines by promoting actin polymerization via its effector protein p140Cap (*Yang et al., 2021*; *Yang et al., 2018*; *Yang et al., 2015*).

Previous studies reported that the double knockout (KO) mice of EndoA1 and EndoA2 (gene name *sh3gl1*) exhibit spontaneous epileptic seizures (*Milosevic et al., 2011*). However, EndoA1 depletion only causes a slight decrease in excitatory synaptic transmission in hippocampal CA1 neurons (*Yang et al., 2021*; *Yang et al., 2018*). As E/I imbalance is caused by changes in either excitatory or inhibitory activity, or both, we were prompted to investigate the biological function of endophilin A1 in the inhibitory synapses. In this study, we show that endophilin A1 interacts with gephyrin and localizes to GABAergic postsynapses. Via its membrane-binding and actin polymerization-promoting activities, endophilin A1 facilitates organization of the inhibitory postsynaptic machinery. Loss of endophilin A1 causes impairment of inhibitory synaptic transmission, E/I imbalance, and aggravation of epileptic susceptibility.

## Results

### Ablation of endophilin A1 causes E/I imbalance and increases epilepsy susceptibility

Previous studies found that EndoA1 and EndoA2 double knockout (KO) mice exhibit spontaneous epileptic seizures (*Milosevic et al., 2011*), suggesting that endophilin A1 and A2 are involved in maintaining the E/I balance. However, another study reported that the protein levels of endophilin A1 were higher in temporal neocortex neurons of temporal lobe epilepsy (TLE) patients (*Yu et al., 2018b*). Moreover, in a chemical kindling mouse model of epilepsy, in which mice are chronically induced with the GABA$_A$R antagonist pentylenetetrazole (PTZ) (*Dhir, 2012*), endophilin A1 was also upregulated in the hippocampus and adjacent temporal cortex, and silencing its expression in the hippocampal CA3 region reduced PTZ-induced seizure susceptibility and severity (*Yu et al., 2018b*). To investigate the role of endophilin A1 in epilepsy, we determined the impact of *sh3gl2* gene ablation on chemical kindling of pan-neural EndoA1 KO mice (*Yang et al., 2018*) with PTZ at a sub-convulsive dose of 25 mg/kg. Compared with the wild-type, PTZ-induced epileptic behaviors were significantly exacerbated in EndoA1 KO mice (*Figure 1A–C* and *Figure 1—video 1*).

We previously found that EndoA1 KO caused a slight decrease in the amplitude of evoked excitatory postsynaptic currents (eEPSC) and a significant attenuation of long-term potentiation (*Yang et al., 2018*). As an imbalance between excitatory and inhibitory signals in neuronal circuits is implicated in epilepsy, we then determined whether loss of endophilin A1 in excitatory neurons affects the E/I ratio by electrophysiological analysis. To this end, we eliminated the *sh3gl2* gene in pyramidal neurons by stereotaxic injection of adeno-associated virus (AAV) expressing *CaMK2α* promoter-driven Cre recombinase into the hippocampal CA1 region of *Sh3gl2$^{fl/fl}$* mice (*Figure 1D*). Whole-cell patch clamp recording of CA1 pyramidal cells indicated that, while the eEPSC and its paired-pulse ratio (PPR) were not significantly changed, the evoked inhibitory postsynaptic current (eIPSC), not its PPR, was decreased in the EndoA1 KO neurons, resulting in a significant increase in the E/I ratio (*Figure 1E–I*). Of note, neither the frequency nor the amplitude of miniature EPSCs (mEPSC) or miniature IPSCs (mIPSC) was altered in EndoA1 KO neurons (*Figure 1J–O*). Moreover, neither fast mIPSCs (perisomatic events:<2.8 ms) nor slow mIPSCs (distal dendritic events:>2.8 ms) (*Miles et al., 1996*) were affected by EndoA1 KO (*Figure 1—figure supplement 1*), indicating that presynaptic quantal release of soma- or dendrite-targeting inhibitory synapses is normal. Taken together, these data indicate that endophilin A1 in hippocampal pyramidal neurons functions in inhibitory synaptic transmission to maintain E/I balance.

### Endophilin A1 is required for the formation and stabilization of inhibitory synapses

Diminished eIPSC in EndoA1 KO pyramidal neurons prompted us to determine the role of endophilin A1 in inhibitory synapses by immunofluorescence staining of brain sections from *Sh3gl2$^{-/-}$* mice. Analysis of confocal microscopy images revealed that, compared with the wild-type, there was a significant decrease in fluorescent signals of both VGAT and gephyrin, the inhibitory pre- and post-synaptic markers, or VGLUT1 and PSD95, the excitatory pre- and postsynaptic markers, in all subregions of the hippocampus of the *Sh3gl2$^{-/-}$* mice on postnatal day 21 (P21) (*Figure 2A, C, D, F*, *Figure 2—figure supplement 1A, C, D, F*), suggesting defects in synapse development. While a slight reduction in excitatory presynaptic protein was observed, the decreases in inhibitory but not excitatory postsynaptic proteins were also detected in the *Sh3gl2$^{-/-}$* mice at P60 (*Figure 2B, C, E, F*, *Figure 2—figure supplement 1B, C, E, F*), indicating that the assembly and/or stabilization of inhibitory postsynapses was impaired throughout brain development. Notably, gephyrin signals in all layers of the different hippocampal subregions were reduced in the *Sh3gl2$^{-/-}$* mice (*Figure 2—figure supplement 1G and H*). In line with the data from immunofluorescent labeling (*Figure 2*, *Figure 2—figure supplement 1*), decreased endogenous PSD95 or gephyrin clusters at synapses were also visualized by the recombinant probes, PSD95.FingR-eGFP or mRuby2-gephyrin.FingR, respectively (*Bensussen et al., 2020*; *Gross et al., 2013*), in EndoA1 KO neurons in the CA1 region at P21 (*Figure 2—figure supplement 2A–D*). As different types of interneurons synapse with hippocampal excitatory neurons at distinct subcellular compartments (soma, axon initial segment, proximal and distal dendrites) indicated by the layered structures of the hippocampus (*Spruston, 2008*), these data suggest that endophilin A1

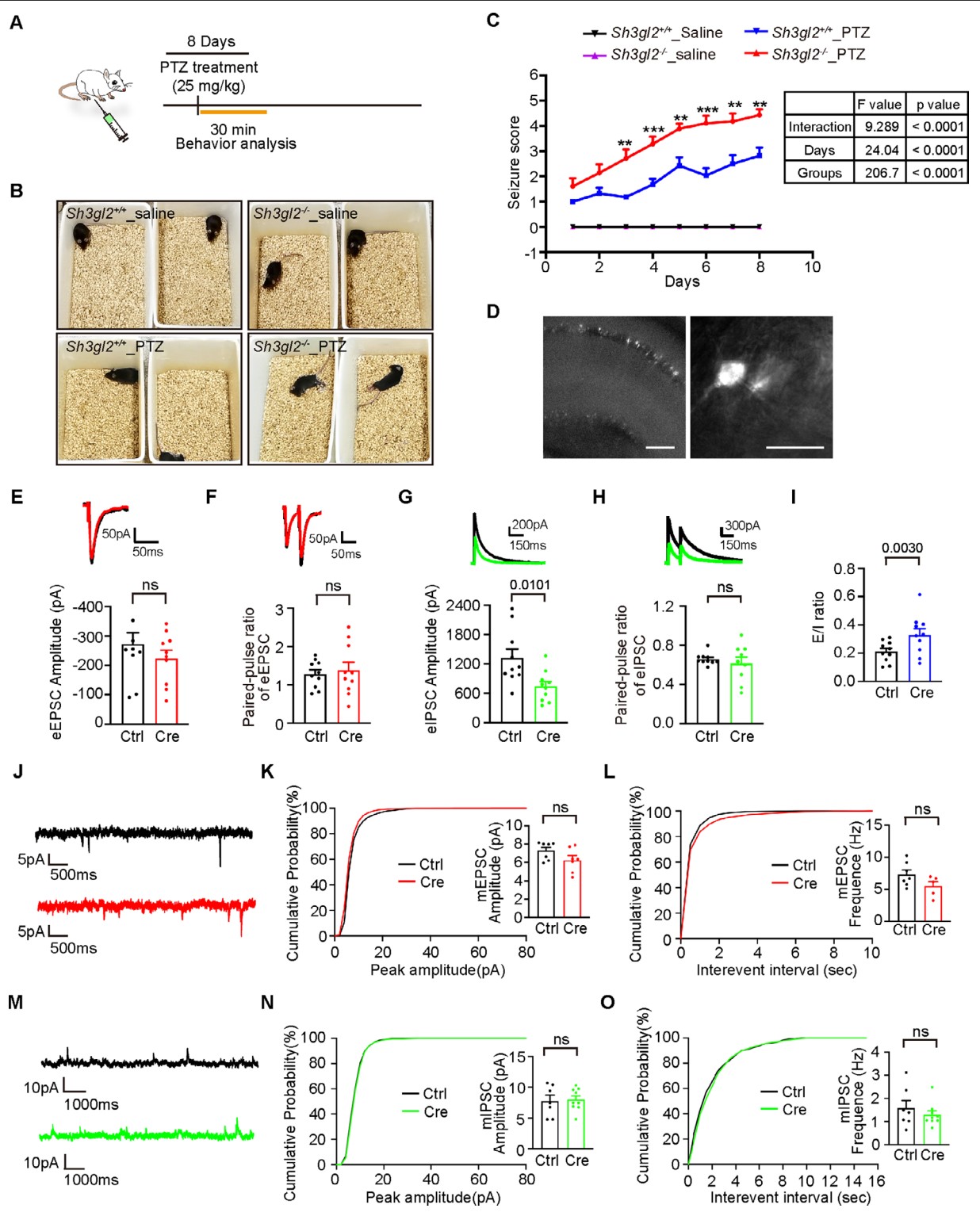

**Figure 1.** Ablation of endophilin A1 in CA1 neurons disrupts the E/I balance and increases epilepsy susceptibility. (**A**) Experimental protocol of PTZ kindling. Saline serves as a negative control for PTZ. (**B**) Representative images of convulsive behavior in each group. (**C**) Seizures were monitored and scored after each injection. Nine 10-week-old mice were used in each condition. Seizure scores are shown as means ± S.E.M. Statistical test: repeated measures ANOVA followed by Tukey post-hoc test at each time point. * p<0.05, ** p<0.01, *** p<0.001, *Sh3gl2⁻/⁻*_PTZ *vs Sh3gl2⁺/⁺*_PTZ at each time point. (**D**) The representative mCherry-positive hippocampal CA1 neurons used for electrophysiological recording. Scale bar, 200 μm or 40 μm. (**E–F**) Representative traces of evoked excitatory postsynaptic currents (eEPSCs) and quantification of mean eEPSC amplitude (**E**), and paired-pulse ratio

*Figure 1 continued on next page*

*Figure 1 continued*

(**F**) from 10 control (Ctrl, uninfected) and 10 EndoA1 KO (mCherry and Cre) pyramidal neurons of four independent animals. (**G–H**) Representative traces of evoked inhibitory postsynaptic currents (eIPSCs) and quantification of mean eIPSC amplitude (**G**), and paired-pulse ratio (**H**) from nine control and nine EndoA1 KO pyramidal neurons of four independent animals. (**I**) Effect of EndoA1 KO on the E/I ratio from nine control and nine EndoA1 KO pyramidal neurons of four independent animals. (**J**) Representative recordings of miniature excitatory postsynaptic currents (mEPSCs) traces (+10 mV) from control or EndoA1 KO CA1 pyramidal cells. (**K**) Cumulative probability and quantification of mEPSC amplitude from seven control and seven EndoA1 KO pyramidal neurons of three independent animals. (**L**) Cumulative probability and quantification of mEPSC frequency from seven control and seven EndoA1 KO pyramidal neurons of three independent animals. (**M**) Representative recordings of miniature inhibitory postsynaptic currents (mIPSCs) traces (–70 mV) from control or EndoA1 KO CA1 pyramidal cells. (**N**) Cumulative probability and quantification of mIPSC amplitude from seven control and seven EndoA1 KO pyramidal neurons of three independent animals. (**O**) Cumulative probability and quantification of mIPSC frequency from seven control and seven EndoA1 KO pyramidal neurons of three independent animals. Statistical test: two-tailed Student's t-test in E-I & K, L, N, O. ns, non-significant. All data shown are scatterplots with means ± S.E.M.

The online version of this article includes the following video, source data, and figure supplement(s) for figure 1:

**Source data 1.** Original file for analysis displayed in *Figure 1C, E-I, K-L & N-O*.

**Figure supplement 1.** Loss of endophilin A1 in vivo does not affect fast mIPSCs (perisomatic events:<2.8 ms) and slow mIPSCs (distal dendritic events:>2.8 ms).

**Figure supplement 1—source data 1.** Original file for analysis displayed in *Figure 1—figure supplement 1*.

**Figure 1—video 1.** PTZ kindling behaviors of *Sh3gl2*[+/+] and *Sh3gl2*[-/-] mice.

https://elifesciences.org/articles/102792/figures#fig1video1

functions in the assembly/stabilization of inhibitory synapses formed between various presynaptic neurons and postsynaptic neurons.

Next, we analyzed the GABA$_A$ receptors and found that, consistent with changes in the inhibitory postsynaptic marker, signals of the GABA$_A$R γ2 subunit were decreased in the hippocampi of both juvenile and adult *Sh3gl2*[-/-] mice (*Figure 2G–I*). These data collectively indicate that endophilin A1 orchestrates the assembly of inhibitory synapses, the architecture of inhibitory postsynapses in particular, and functioning of GABAergic synapses.

## Endophilin A1 localizes to the iPSD and augments the organization of the inhibitory postsynaptic machinery

To substantiate the role of postsynaptic endophilin A1 in the assembly of inhibitory synapses and exclude the impacts of presynaptic endophilin A1 and its depletion on early embryonic development, we determined the effects of endophilin A1 KO on inhibitory and excitatory synapses in cultured hippocampal pyramidal neurons during late developmental stages. As the expression levels of GABA$_A$R α1 are highly heterogeneous in individual cultured neurons (*Figure 3—figure supplement 1A*), we determined the effect of Cre-mediated EndoA1 KO on surface expression of GABA$_A$R γ2 in *Sh3gl2*[fl/fl] hippocampal neurons by immunofluorescence staining. While no changes in surface GluA1 was detected, the fluorescence signal intensity and number of surface GABA$_A$R γ2 clusters were largely lower in dendrites of EndoA1 KO neurons, which was rescued by re-expression of endophilin A1 (*Figure 3A–D*). We further examined the total and surface levels of synaptic proteins by immunoblotting. Biotinylation assay of surface proteins revealed that, compared with wild-type, although there was no change in total levels of synaptic proteins in EndoA1 KO pyramidal neurons, the surface expression of GABA$_A$Rs but not NL2 or the NR1 subunit of the NMDA-type glutamate receptor was significantly reduced (*Figure 3E and F*). Collectively, these data indicate that postsynaptic endophilin A1 enhances the surface clustering of GABA$_A$Rs and facilitates the assembly of inhibitory postsynapses in pyramidal neurons.

Next, we sought to determine the molecular function of endophilin A1 in inhibitory synapses at the cellular level. By immunofluorescence staining and super-resolution microscopy of hippocampal neurons in dissociated culture, we observed partial colocalization of endophilin A1 with gephyrin as well as surface GABA$_A$R γ2 in dendrites of hippocampal neurons, and juxtaposition or partial colocalization of endophilin A1 and VGAT puncta (*Figure 3G–I* and *Figure 3—video 1–3*). We further detected the localization of endophilin A1 to inhibitory synapses by co-immunostaining with both pre- and post-synaptic markers (*Figure 3—figure supplement 1B* and *Figure 3—video 4*). Quantitative analysis of super-resolution localization maps revealed that ~47% puncta of gephyrin or Bassoon were proximal to endophilin A1 (*Figure 3—figure supplement 1G*, n = 14), with a mean distance between

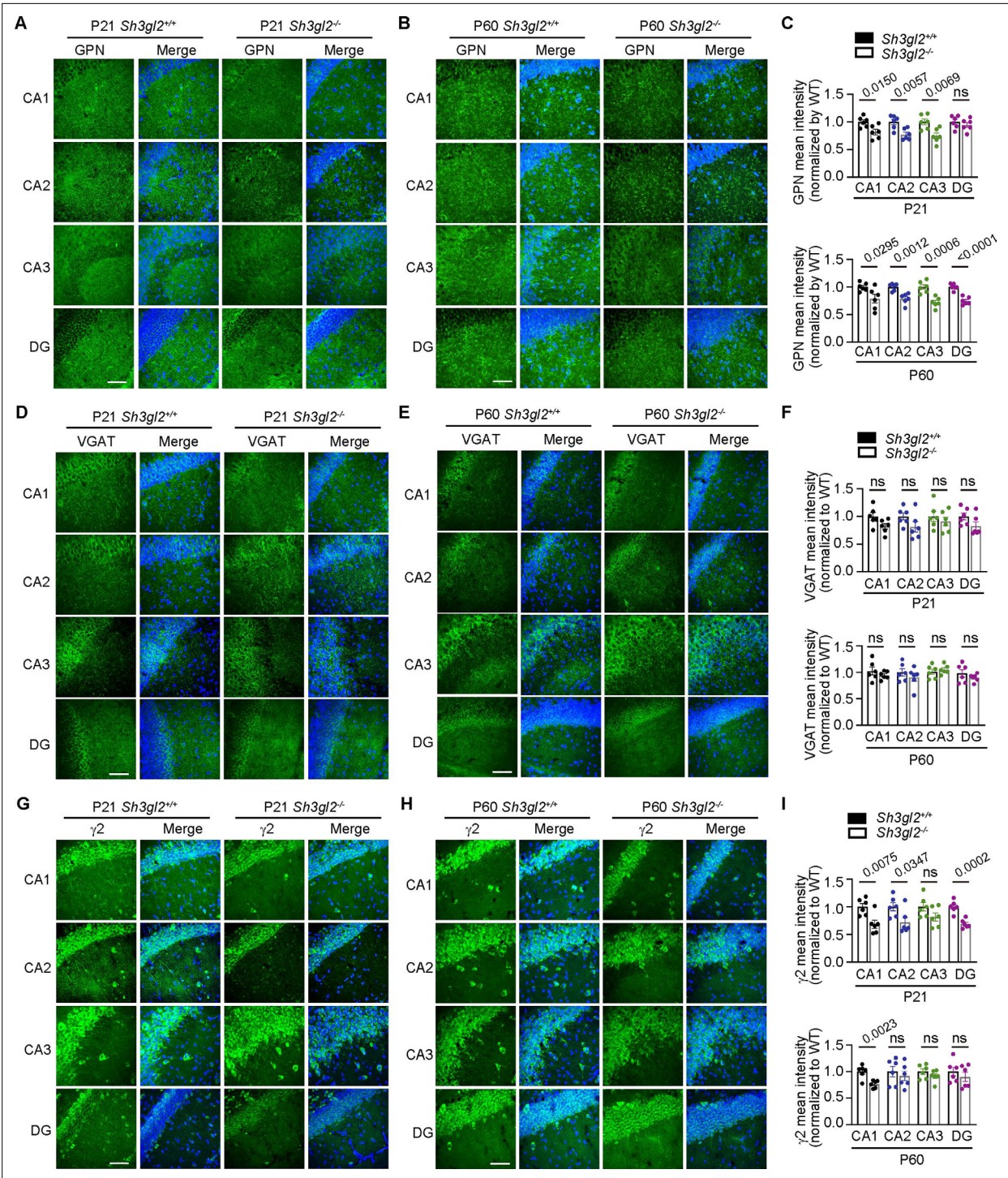

**Figure 2.** EndoA1 KO causes defects in the formation and stabilization of inhibitory synapses. (**A–B**) Immunofluorescence staining and confocal microscopy of gephyrin (GPN) in the hippocampal region of 3-week-old (P21) and 10-week-old (P60) *Sh3gl2+/+* or *Sh3gl2-/-* mouse brains. Scale bar, 50 µm. (**C**) Quantification of the signal intensity of GPN in (**A**) and (**B**). n = 6 mice for each genotype. (**D–E**) Same as (**A–B**) except that slices were immunostained for VGAT, (**D**) for P21, (**E**) for P60. Scale bar, 50 µm. (**F**) Quantification of VGAT signal intensity in (**D**) and (**E**). n = 6 mice for each genotype. (**G–H**) Same as (**A–B**) except that slices were immunostained for GABA$_A$R γ2 (γ2). (**G**) for P21, (**H**) for P60. Scale bar, 50 µm. (**I**) Quantification of γ2 signal intensity in (**G**) and (**H**). n = 6 mice for each genotype. Statistical test: unpaired two-tailed Student's t-test. ns, non-significant. All data are scatterplots with means ± S.E.M. Abbreviations: CA1-3, cornus ammonis region 1–3; DG, dentate gyrus; DAPI staining of cell nuclei is shown in blue.

The online version of this article includes the following source data and figure supplement(s) for figure 2:

**Source data 1.** Original file for analysis displayed in *Figure 2C*.

*Figure 2 continued on next page*

*Figure 2 continued*

**Source data 2.** Original file for analysis displayed in *Figure 2F*.

**Source data 3.** Original file for analysis displayed in *Figure 2I*.

**Figure supplement 1.** Effects of EndoA1 knockout on excitatory and inhibitory synapses in the hippocampus.

**Figure supplement 1—source data 1.** Original file for analysis displayed in *Figure 2—figure supplement 1C*.

**Figure supplement 1—source data 2.** Original file for analysis displayed in *Figure 2—figure supplement 1F*.

**Figure supplement 1—source data 3.** Original file for analysis displayed in *Figure 2—figure supplement 1H*.

**Figure supplement 2.** Changes of endogenous postsynaptic proteins visualized by recombinant probes after depletion of EndoA1.

**Figure supplement 2—source data 1.** Original file for analysis displayed in *Figure 2—figure supplement 2B and D*.

endophilin A1- and gephyrin-positive pixels of ~ 120 nm, or between endophilin A1- and Bassoon-positive pixels of ~ 130 nm (*Figure 3—figure supplement 1C–F*). These results indicate that endophilin A1 localizes to not only ePSD but also iPSD. Therefore, we speculated that endophilin A1 acted as a component of iPSD. Previous studies reported that synaptic GABA$_A$Rs contain the γ2 subunit along with α and β subunits (e.g. α1β2γ2) (*Scott and Aricescu, 2019*). The γ2 subunit is essential for receptor clustering at the synapse, mediated by interactions with scaffold proteins (*Essrich et al., 1998*). Indeed, co-immunoprecipitation (coIP) of endogenous proteins in membrane fractions from mouse brain lysates with antibodies against GABA$_A$R α1 detected not only the known iPSD components γ2, NL2, and gephyrin, but also endophilin A1, and vice versa (*Figure 3J*). Of note, loss of endophilin A1 caused decreases in levels of GABA$_A$R γ2 and gephyrin but not NL2 in anti-GABA$_A$R α1 immunoprecipitates (*Figure 3K*). Further, immunoblotting of synaptosome fractions revealed that the levels of GABA$_A$R γ2 and gephyrin were reduced at synapses in adult EndoA1 KO mice (*Figure 3L*). Together, these data support that endophilin A1 functions at the iPSD to promote the assembly and/or stabilization of the GABAergic postsynaptic machinery.

## Endophilin A1 directly interacts with the scaffold protein gephyrin and promotes its clustering and localization of GABA$_A$Rs to iPSD

We subsequently sought to investigate the mechanistic role of endophilin A1 in the clustering of inhibitory postsynaptic proteins. To this end, we performed coIP from mouse brain lysates with antibodies against endophilin A1 (*Figure 4A*) and analyzed the immunoprecipitates by mass spectrometry. Gene Ontology (GO) analysis revealed that, in addition to proteins participating in known cellular processes involving endophilin A1, such as synaptic vesicle recycling, actin cytoskeleton remodeling, endocytosis, and long-term potentiation (*Gad et al., 2000*; *Ringstad et al., 1999*; *Schuske et al., 2003*; *Verstreken et al., 2003*; *Yang et al., 2021*; *Yang et al., 2018*; *Yang et al., 2015*), proteins related to not only glutamatergic but also dopaminergic and GABAergic synapses were also identified, including gephyrin (*Figure 4B* and *Supplementary file 1*).

Gephyrin is the major scaffold protein that interacts with the intracellular domains of GABA$_A$Rs and anchors them at the inhibitory postsynaptic membrane (*Essrich et al., 1998*; *Kneussel et al., 1999*; *Mukherjee et al., 2011*; *Tretter et al., 2008*). The cell adhesion molecule NL2 drives postsynaptic assembly at perisomatic inhibitory synapses via interaction with gephyrin and the membrane lipid-binding protein collybistin (*Poulopoulos et al., 2009*). We speculated that endophilin A1 functions in the assembly of iPSD via interaction with the scaffold and/or intercellular adhesion proteins. To determine which inhibitory synaptic protein(s) interacts with endophilin A1, we performed coIP from HEK293T cells transiently expressing endophilin A1 and NL2, or gephyrin, and detected interaction of endophilin A1 with gephyrin but not NL2 (*Figure 4C and D*). Moreover, coIP from mouse brain lysates corroborated interaction between endophilin A1 and gephyrin but not GABA$_A$R α1 or γ2 (*Figure 4E*). Further, neither endophilin A2 nor A3 interacted with gephyrin (*Figure 4F*).

Notably, although the expression levels of gephyrin were not affected by ablation of *Sh3gl2*, its distribution in the PSD fraction was decreased (*Figure 3L*). In dendrites of mature cultured hippocampal neurons following *Sh3gl2* depletion, the signal intensity and number of gephyrin.FingR-eGFP clusters were also significantly attenuated, with no obvious changes in PSD95.FingR-eGFP clusters (*Figure 4—figure supplement 1A–D*), which was consistent with the reduced gephyrin but unaltered PSD95 labeled by antibodies in brain slice of mature EndoA1 KO mice (*Figure 2*, *Figure 2—figure*

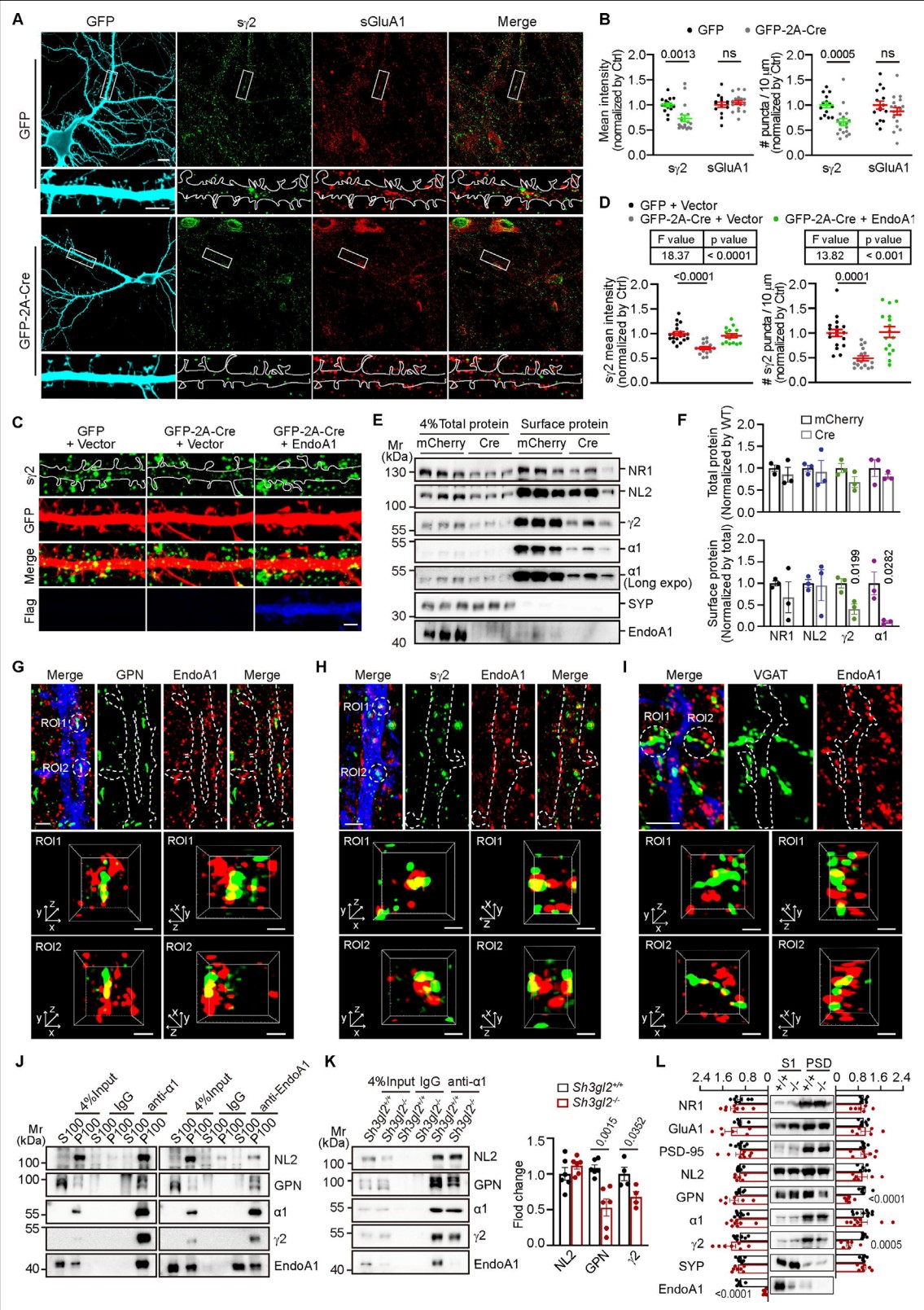

**Figure 3.** Endophilin A1 is distributed in the iPSD and regulates the assembly of the inhibitory postsynaptic machinery. (**A**) Cultured *Sh3gl2^fl/fl* hippocampal neurons were transfected with a construct expressing GFP or GFP-2A-Cre on DIV12, fixed, and immunostained for surface GluA1 (sGluA1) or GABA_AR γ2 (sγ2) on DIV16. Shown are representative confocal microscopy images. Lower panels are magnifications of boxed regions in the upper panels. Scale bar, 10 μm in the upper panels and 5 μm in the lower panels. (**B**) Quantification of the signal intensity (left) and cluster number (right) in

*Figure 3 continued on next page*

*Figure 3 continued*

dendrites in (**A**). GFP: 15 neurons; Cre: 18 neurons, from 3 independent cultures. Statistical test: unpaired two-tailed Student's t-test. ns, non-significant. (**C–D**) Cultured *Sh3gl2*^fl/fl^ hippocampal neurons were co-transfected with constructs for GFP or GFP-2A-Cre and Flag vector or Flag-endophilin A1 (Flag-EndoA1) on DIV12, and immunostained for surface γ2 on DIV16. Shown are representative confocal images (**C**) and quantification of surface γ2 in dendrites (**D**). Scale bar, 2 µm. GFP: 18 neurons; Cre: 19 neurons; Cre + EndoA1: 18 neurons, from 3 independent cultures. Statistical test: one-way ANOVA by Tukey post hoc test. (**E–F**) *Sh3gl2*^fl/fl^ hippocampal neurons were infected with adeno-associated virus (AAV) carrying mCherry or mCherry-2A-Cre on DIV3 and surface biotinylated on DIV16. Total and surface proteins were detected by SDS-PAGE and immunoblotting. N = 3. Statistical test: unpaired two-tailed t-test. (**G**) Hippocampal neurons were transfected with a construct expressing GFP to label neuronal morphology on DIV12, then immunostained with antibodies against EndoA1 and GPN on DIV16. Shown are representative z-stack maximum projection images captured by structured illumination microscopy (SIM). Lower panels are insets of boxed regions in the upper panel. Dendrites and spines are outlined in white. Scale bars, 2 µm in upper panel and 0.5 µm in magnified images. (**H**) Same as (**G**) except that neurons were immunostained for EndoA1 and surface γ2. (**I**) Same as (**G**) except that neurons were immunostained for EndoA1 and VGAT. (**J**) Immunoisolation from the membrane fraction of mouse brain tissues with antibodies against GABA_AR α1 (α1) or EndoA1. (**K**) Immunoisolation from *Sh3gl2*^+/+^ and *Sh3gl2*^-/-^ brain lysates with antibodies to GABA_AR α1. Shown are representative immunoblot (left) and quantification of neuroligin 2 (NL2), GPN, and GABA_AR γ2 co-immunoisolated with α1 (right). N = 6 or 4 independent experiments. Statistical test: unpaired two-tailed Student's t-test. (**L**) PSD fractions were isolated by differential centrifugation from mouse brain lysates and analyzed by immunoblotting with antibodies to EndoA1 and synaptic proteins. S1, homogenates; PSD, postsynaptic density. SYP, synaptophysin. N = 7 independent experiments. Statistical test: unpaired two-tailed Student's t-test. All data represent scatter plots with means ± S.E.M.

The online version of this article includes the following video, source data, and figure supplement(s) for figure 3:

**Source data 1.** PDF file containing original western blots with the relevant bands clearly labeled displayed in *Figure 3E, G and K & L* .

**Source data 2.** The original files of the full raw uncropped, unedited blots displayed in *Figure 3E, G and K & L*.

**Source data 3.** Original file for analysis displayed in *Figure 3B*.

**Source data 4.** Original file for analysis displayed in *Figure 3D*.

**Source data 5.** Original file for analysis displayed in *Figure 3F & K*.

**Source data 6.** Original file for analysis displayed in *Figure 3L*.

**Figure supplement 1.** Expression of GABA_AR α1 in cultured hippocampal neurons and the localization of EndoA1 in inhibitory pre- and postsynaptic sites.

**Figure supplement 1—source data 1.** Original file for analysis displayed in *Figure 3—figure supplement 1G*.

**Figure 3—video 1.** The spatial relationship between EndoA1 and GPN.

https://elifesciences.org/articles/102792/figures#fig3video1

**Figure 3—video 2.** The spatial relationship between endophilin A1 and surface GABA_AR γ2.

https://elifesciences.org/articles/102792/figures#fig3video2

**Figure 3—video 3.** The spatial relationship between endophilin A1 and VGAT.

https://elifesciences.org/articles/102792/figures#fig3video3

**Figure 3—video 4.** The spatial relationship between Bassoon, gephyrin and endophilin A1.

https://elifesciences.org/articles/102792/figures#fig3video4

*supplement 1*), Moreover, immunofluorescence staining revealed a decrease in the clustering of gephyrin across the soma, the axon initial segment (AIS), and dendrites (*Figure 4—figure supplement 1E–H*). The gephyrin clustering phenotype was rescued by expression of endophilin A1 but not endophilin A2 or A3 (*Figure 4G and H*), suggesting that their functions at the inhibitory postsynapses are not redundant. Conversely, overexpression of endophilin A1 caused an increase in the mean signal intensity and number of gephyrin puncta in the soma and dendrites (*Figure 4I and J*). Collectively, these data demonstrate that endophilin A1 interacts with the scaffold protein gephyrin and facilitates its clustering at inhibitory postsynapses.

At the submembranous postsynaptic density, gephyrin self-assembles via the N-terminal G and C-terminal E domains to provide scaffold for inhibitory postsynapses and anchors GABA_ARs to the postsynaptic sites. Consistent with the effect of EndoA1 KO on the clustering of gephyrin (*Figure 4G and H*) and GABA_AR γ2 (*Figure 3A–D*), immunofluorescence staining and confocal microscopy analysis showed that depletion of endophilin A1 also caused a decrease in the localization of GABA_AR γ2 to gephyrin puncta in EndoA1 KO neurons (*Figure 5A–C*). Moreover, overexpression of endophilin A1 also enhanced clustering of GABA_AR γ2 in the soma and dendrites (*Figure 5D and E*). Together, these data indicate that endophilin A1 promotes anchoring of GABA_ARs to the iPSD.

To further understand the mechanism by which endophilin A1 regulates the assembly and function of inhibitory postsynapses, next we attempted domain mapping of the endophilin A1-gephyrin

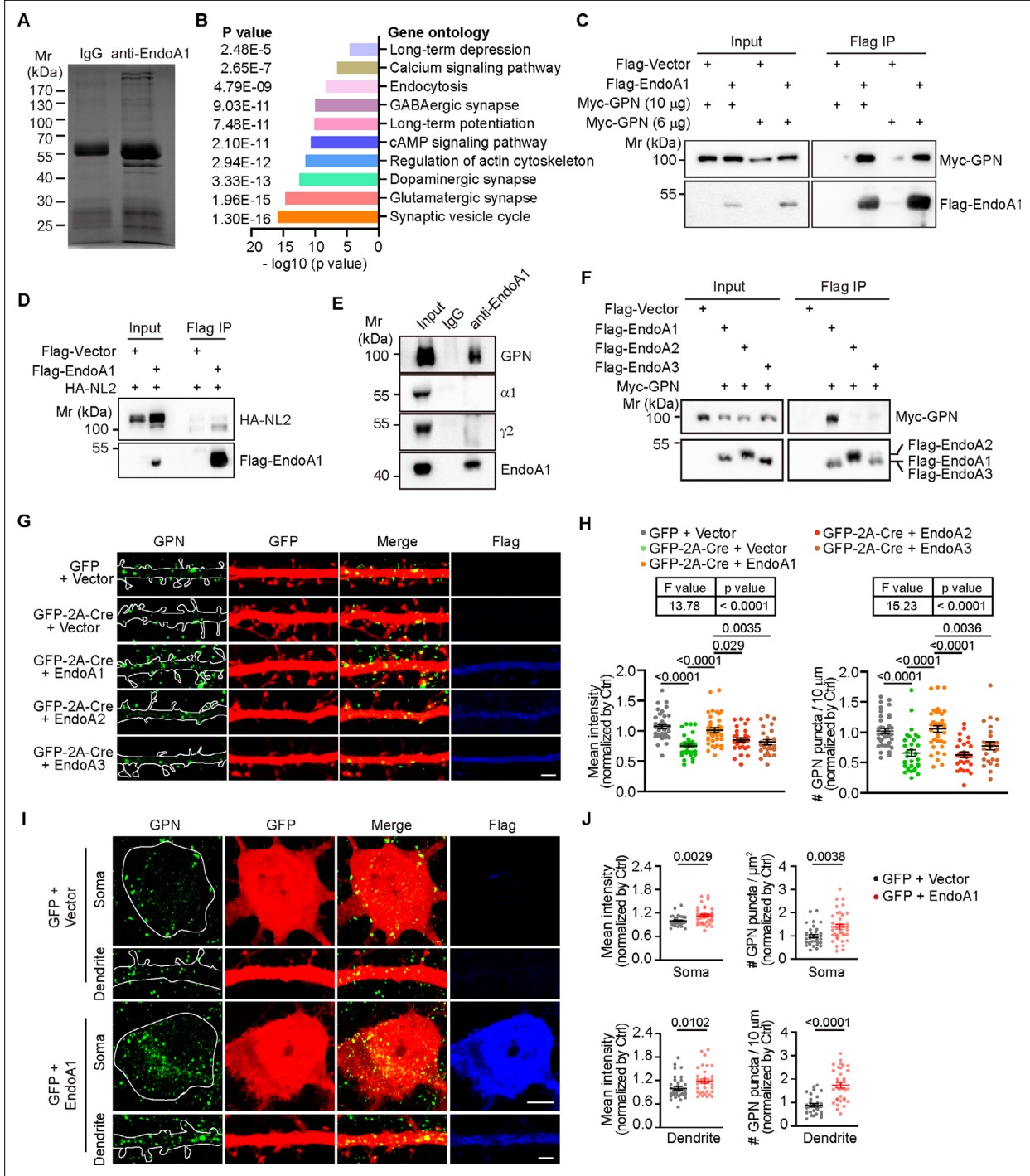

**Figure 4.** Endophilin A1 interacts with gephyrin and promotes its clustering in postsynaptic neurons. (**A**) Silver staining of proteins co-immunoprecipitated from *Sh3gl2*[+/+] mouse brain lysates with antibodies to endophilin A1. (**B**) Gene Ontology (GO) enrichment analysis of EndoA1-interacting proteins identified by mass spectrometry. (**C–D**) HEK293T cells co-transfected with constructs expressing Myc-tagged GPN or HA-tagged NL2 and Flag-tagged EndoA1 for 48 hr were lysed for Flag IP. Input and bound proteins were analyzed by immunoblotting with antibodies against Myc, HA, and Flag. (**E**) CoIP from brain lysates with antibodies to EndoA1. (**F**) HEK293T cells overexpressing Flag-tagged endophilin As and Myc-tagged GPN were lysed for co-immunoprecipitation. (**G**) *Sh3gl2*[fl/fl] hippocampal neurons were co-transfected with constructs expressing GFP and Flag vector, or GFP-2A-Cre and Flag vector, Flag-EndoA1, Flag-EndoA2, or Flag-EndoA3 on DIV12, fixed on DIV16, and immunostained for GPN, GFP, and Flag. Shown are representative confocal microscopy images. Scale bar, 2 µm. (**H**) Quantification of GPN signal intensity (left) and cluster number (right) in dendrites in (**G**). 36 neurons for GFP + vector; 34 neurons for Cre + vector; 38 neurons for Cre + EndoA1; 29 neurons for Cre + EndoA2; 27 neurons for Cre + EndoA3, from 3 independent cultures. Statistical test: one-way ANOVA by Tukey post hoc test. (**I**) Hippocampal neurons were co-transfected

*Figure 4 continued on next page*

*Figure 4 continued*

with constructs expressing GFP and vector or Flag-EndoA1 on DIV12, fixed, and immunostained for GPN on DIV16. Shown are representative confocal microscopy images of GPN in the soma (left) and dendrites (right). Scale bars, 5 μm in the upper panels and 2 μm in the lower panels. (**J**) Quantification of GPN signal intensity (left) and cluster number (right) in the soma and dendrites. 34 neurons for GFP + vector; 37 neurons for GFP + EndoA1, from 3 independent cultures. Statistical test: unpaired Student's two-tailed t-test. All data represent scatterplots with means ± S.E.M.

The online version of this article includes the following source data and figure supplement(s) for figure 4:

**Source data 1.** PDF file containing original western blots with the relevant bands clearly labeled displayed in *Figure 4A, C, D and E & F*.

**Source data 2.** The original files of the full raw uncropped, unedited gels or blots displayed in *Figure 4A, C, D and E & F*.

**Source data 3.** Original file for analysis displayed in *Figure 4B*.

**Source data 4.** Original file for analysis displayed in *Figure 4H*.

**Source data 5.** Original file for analysis displayed in *Figure 4J*.

**Figure supplement 1.** Effect of EndoA1 knockout on gephyrin clustering in cultured hippocampal neurons.

**Figure supplement 1—source data 1.** Original file for analysis displayed in *Figure 4—figure supplement 1B and D*.

**Figure supplement 1—source data 2.** Original file for analysis displayed in *Figure 4—figure supplement 1H*.

interaction. To this end, we generated a series of truncation and internal deletion mutants for endophilin A1 or gephyrin (*Figure 5F*). For those that did not express in mammalian cells, we determined their interaction with gephyrin by in vitro binding assays with recombinant proteins. None of the truncation or internal deletion mutants lost the ability to bind to gephyrin, and deletion of neither the BAR nor the SH3 domain weakened the binding (*Figure 5G and H*), suggesting that they are involved in the interaction with gephyrin. Pull-down assay showed both G and E domains but not C domain of gephyrin were involved in the interaction with endophilin A1 (*Figure 5I*). Indeed, the decreases in clustering of either gephyrin or GABA$_A$R γ2 in EndoA1 KO neurons were not rescued by expression of endophilin A1 fragments (*Figure 5J–M*), indicating that all functional domains of endophilin A1 are required for the assembly of the GABAergic postsynaptic machinery.

## Actin polymerization is required for stabilization of inhibitory postsynapses

Endophilin A1 functions in the morphogenesis and structural plasticity of dendritic spines by promoting actin polymerization underneath the plasma membrane (*Yang et al., 2021*; *Yang et al., 2018*; *Yang et al., 2015*). Of note, previous studies reported controversial results about the role of the actin and microtubule cytoskeleton in gephyrin clustering (*Allison et al., 2000*; *Charrier et al., 2006*; *Kirsch and Betz, 1995*). To further investigate the mechanism underlying endophilin A1-mediated assembly of inhibitory postsynapses, we asked whether actin polymerization contributes to the stabilization of iPSD and GABA$_A$R clustering in dendrites. Consistent with previous studies (*van Bommel et al., 2019*), we observed large patches of F-actin labeled by LifeAct-mCherry in spines and small puncta in dendritic shafts that exhibited partial colocalization with or adjoined gephyrin in cultured hippocampal neurons by super-resolution microscopy (*Figure 6—figure supplement 1A* and *Figure 6—video 1*). We then determined the impact of F-actin depolymerization on gephyrin-labeled iPSD and γ2-containing GABA$_A$Rs in cultured hippocampal neurons. Treatment of mature neurons with nocodazole, a microtubule depolymerizing reagent, for one hour (short-term) or two hours (long-term), caused decreases in the number of both PSD95 and gephyrin puncta (*Figure 6A–D*), which was in good agreement with previous findings that microtubule dynamics regulates the spatial organization of both excitatory and inhibitory postsynaptic structures (*Gu et al., 2008*; *Jaworski et al., 2009*; *Kirsch and Betz, 1995*; *Kirsch et al., 1991*). Short-term or long-term incubation with latrunculin A, which disrupts actin polymerization, also caused decreases in the number of both ePSD and iPSD puncta (*Figure 6A–D*), indicating that actin polymerization also contributes to the stabilization of the iPSD. We then tested the effect of actin depolymerization on GABA$_A$R clustering and found that long-term but not short-term latrunculin A application caused a reduction in the number of GABA$_A$R γ2 puncta in dendrites (*Figure 6E–H*). This finding supported the observation that the decrease in gephyrin cluster preceded the reduction in GABA$_A$R γ2 receptor cluster following latrunculin A treatment (*Figure 6I*). In addition, colocalization analysis revealed less recruitment of γ2 to gephyrin puncta even after short-term drug application, without changes in γ2 clustering (*Figure 6J and K*). Together, these

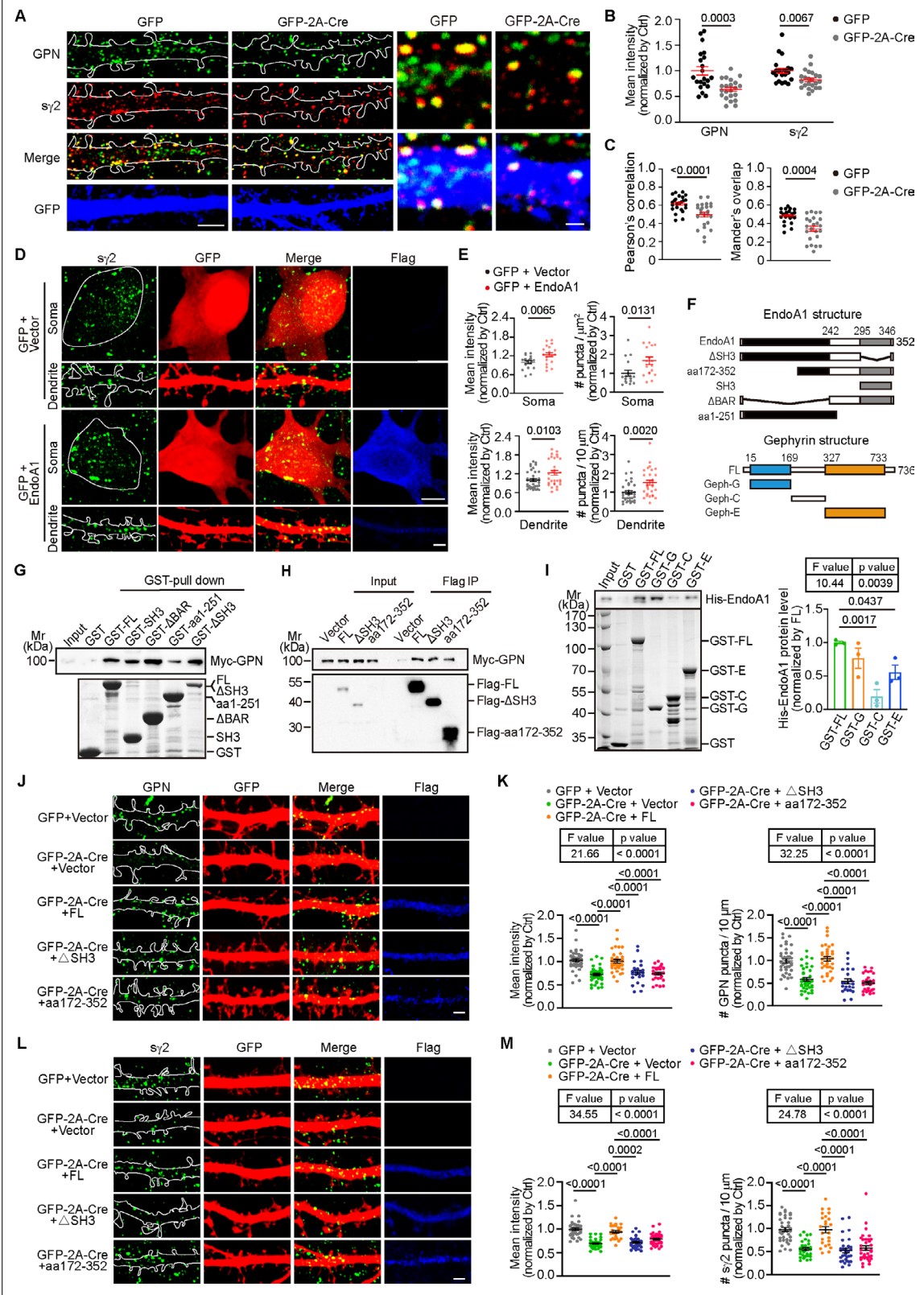

**Figure 5.** Both BAR and SH3 domains are required for endophilin A1-mediated organization of iPSD. (**A**) *Sh3gl2*$^{fl/fl}$ hippocampal neurons expressing GFP or Cre were immunostained for GPN, surface γ2, and GFP on DIV16. Shown are representative confocal microscopy images. Scale bars, 5 μm in left panels and 1 μm in magnified images. (**B–C**) Quantification of the mean intensity of GPN (green) and surface γ2 (red) and their colocalization in (**A**). 21 neurons for GFP, 22 neurons for Cre, from 3 independent cultures. Statistical test: unpaired two-tailed Student's t-test. All data represent scatterplots

*Figure 5 continued on next page*

*Figure 5 continued*

with means ± S.E.M. (**D**) DIV12 hippocampal neurons were co-transfected with constructs expressing GFP and vector or Flag-EndoA1, fixed on DIV16 and immunostained for surface γ2. Shown are representative confocal microscopy images of surface γ2 in the soma (upper) and dendrites (lower). Scale bars, 5 µm in the upper panels and 2 µm in the lower panels. (**E**) Quantification of surface γ2 mean intensity (left) and cluster number (right) in the soma and dendrites. Soma: 20 neurons for GFP + vector, 20 neurons for GFP + EndoA1; Dendrites: 35 neurons for GFP + vector, 30 neurons for GFP + EndoA1, from 3 independent cultures. Statistical test: unpaired two-tailed Student's t-test. All data represent scatterplots with means ± S.E.M. (**F**) Schematic representation of EndoA1 and GPN domain structure and diagrams of fragments used for interaction mapping. (**G**) GST-tagged EndoA1 fragments conjugated to Glutathione-Sepharose beads were incubated with lysates of HEK293T cells overexpressing Myc-tagged GPN. Input and bound proteins were analyzed by immunoblotting with antibodies against Myc. Bottom panel: Coomassie blue-stained SDS-PAGE gel showing GST fusion proteins. FL, full length. (**H**) HEK293T cells co-transfected with constructs encoding Myc-tagged GPN and Flag-tagged EndoA1 fragments were lysed for Flag IP. Input and bound proteins were analyzed by immunoblotting with antibodies against Myc and Flag. (**I**) Binding of His-tagged EndoA1 to GST-tagged GPN fragments in pull-down assay. Bound proteins were analyzed by SDS-PAGE and immunoblotting with anti-His antibodies. Input and bound proteins were analyzed by immunoblotting with antibodies against His. Bottom panel: Coomassie blue-stained SDS-PAGE gel showing GST fusion proteins. FL, full length. Right panel: Quantification of EndoA1 binding to GPN domains. N = 3 independent experiments. Statistical test: one-way ANOVA by Tukey post hoc test. All data represent scatterplots with means ± S.E.M. (**J**) DIV12 *Sh3gl2^{fl/fl}* hippocampal neurons were co-transfected with constructs expressing GFP and Flag vector or GFP-2A-Cre, and Flag vector or Flag-EndoA1 FL, Flag-EndoA1 ΔSH3, Flag-EndoA1 aa172-352, fixed and immunostained for GPN on DIV16. Shown are representative confocal microscopy images of GPN. Scale bar, 2 µm. (**K**) Quantification of mean intensity (left) and cluster number (right) in dendrites is shown for GPN in (**J**). 41 neurons for GFP + vector; 44 neurons for Cre + vector; 35 neurons for Cre + FL; 24 neurons for Cre + ΔSH3; 27 neurons for Cre + aa172-352, from 3 independent cultures. Statistical test: one-way ANOVA by Tukey post hoc test. All data represent scatterplots with means ± S.E.M. (**L**) Same as (**J**) except that neurons were immunostained for surface γ2. Scale bar, 2 µm. (**M**) Quantification of mean intensity (left) and cluster number (right) in dendrites is shown for surface γ2 in (**L**). 38 neurons for GFP + vector; 31 neurons for Cre + vector; 26 neurons for Cre + FL; 33 neurons for Cre + ΔSH3; 36 neurons for Cre + aa172-352, from 3 independent cultures. Statistical test: one-way ANOVA by Tukey post hoc test. All data represent scatterplots with means ± S.E.M.

The online version of this article includes the following source data for figure 5:

**Source data 1.** PDF file containing original western blots with the relevant bands clearly labeled displayed in *Figure 5G, H, I*.

**Source data 2.** The original files of the full raw uncropped, unedited gels or blots displayed in *Figure 5G, H, I*.

**Source data 3.** Original file for analysis displayed in *Figure 5B, C and I* .

**Source data 4.** Original file for analysis displayed in *Figure 5E*.

**Source data 5.** Original file for analysis displayed in *Figure 5K*.

**Source data 6.** Original file for analysis displayed in *Figure 5M*.

data demonstrate that actin polymerization is essential for the stabilization of iPSD and anchoring of GABA_ARs to the postsynaptic sites.

## Both the membrane-binding and p140Cap-binding capacities of endophilin A1 are indispensable for its function at the inhibitory postsynapses

At the excitatory synapse, the plasma membrane-associated endophilin A1 regulates the morphogenesis, maturation, and structural plasticity of dendritic spines by promoting branched actin polymerization underneath the postsynaptic membrane via its interaction with the cytoskeleton regulator p140Cap (*Yang et al., 2021*; *Yang et al., 2018*; *Yang et al., 2015*). Having established the requirement for both endophilin A1 and the actin cytoskeleton in the assembly and stabilization of the inhibitory postsynaptic structure, we speculated that regulation of inhibitory postsynapse assembly by endophilin A1 necessitates its plasma membrane-binding and actin polymerization-promoting activities. Silencing p140Cap expression in postsynaptic CA1 pyramidal cells by shRNA-mediated knockdown caused decreases in the clustering of gephyrin puncta in dendrites, which was restored by overexpression of p140Cap (*Figure 6—figure supplement 1B, C*), suggesting that p140Cap-mediated actin polymerization is required for the assembly of iPSD in postsynaptic neurons. However, compared with mere depletion of endophilin A1, p140Cap knockdown did not further attenuate the clustering of gephyrin in EndoA1 KO neurons (*Figure 6—figure supplement 1D, E*). Moreover, overexpression of p140Cap did not rescue the iPSD phenotype in EndoA1 KO neurons (*Figure 7A and B*), suggesting that p140Cap-enhanced actin polymerization promotes the organization of iPSD through endophilin A1-mediated pathway. Indeed, overexpression of neither the membrane-binding deficient (KKK-EEE) nor the p140Cap-binding deficient (Y343A) mutant of endophilin A1 (*Yang et al., 2021*) could rescue the iPSD defect (*Figure 7A and B*). Consistently, neither p140Cap nor the endophilin A1

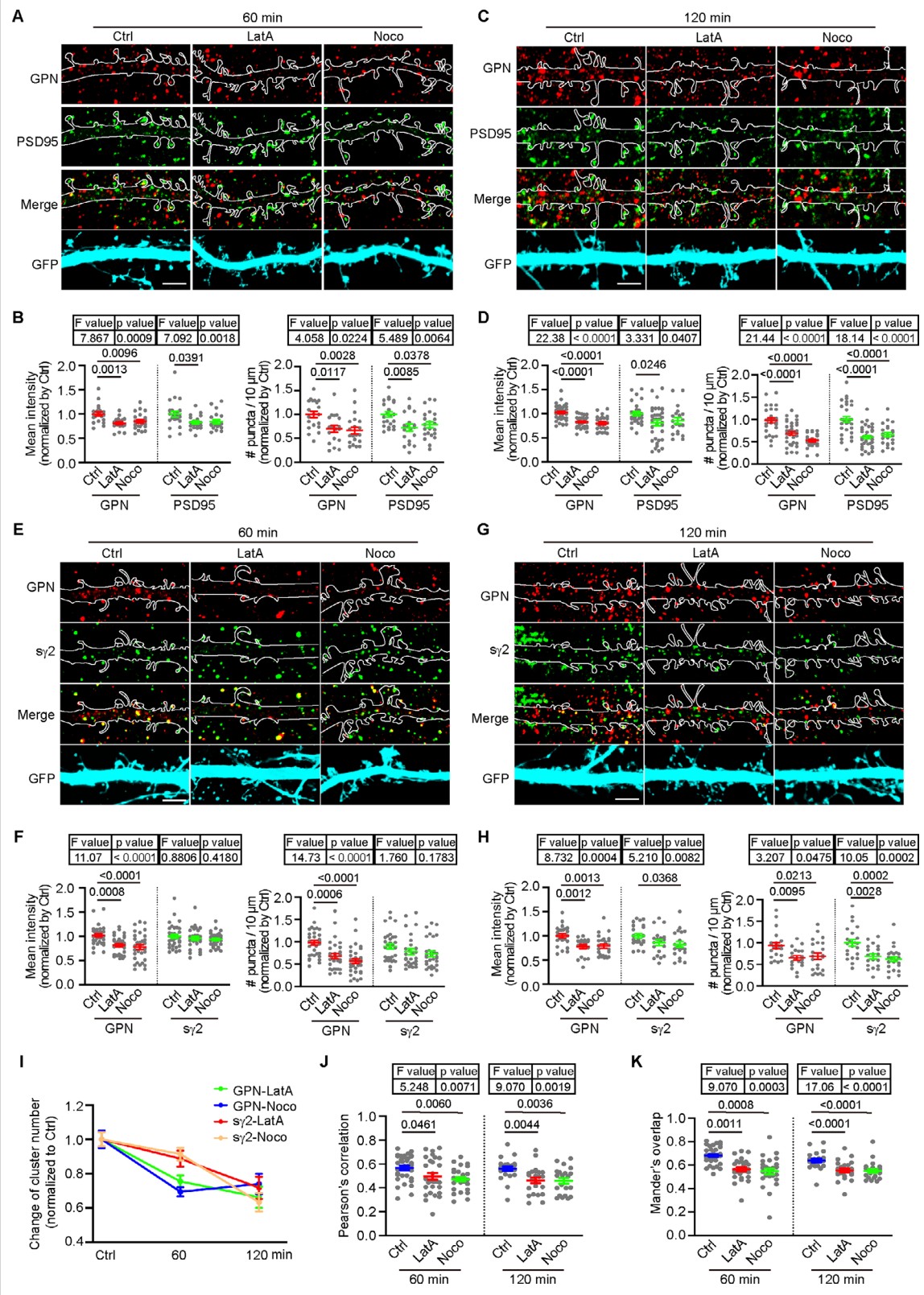

**Figure 6.** Actin polymerization is required for stabilization of inhibitory postsynapses. (**A–D**) DIV12 cultured hippocampal neurons transiently expressing GFP were treated with DMSO (vehicle control), 1 µM Latrunculin A (LatA) or 1 µM Nocodazole (Noco) for 60 min (**A**) or 120 min (**C**) on DIV16, respectively. Changes in postsynaptic proteins GPN (red) and PSD95 (green) were detected by confocal microscopy. Quantitative data shown were the mean intensity (left) and cluster number (right) of GPN and PSD95 at 60 min (**B**) and 120 min (**D**). Ctrl: 20 neurons for 60 min; 29 neurons for 120 min;

*Figure 6 continued on next page*

*Figure 6 continued*

LatA: 20 neurons for 60 min; 33 neurons for 120 min; Noco: 23 neurons for 60 min; 22 neurons for 120 min, from 3 independent cultures. Statistical test: one-way ANOVA by Tukey post hoc test. Scale bar, 5 µm. (**E–H**) Same as (**A–D**) except that neurons were immunostained for GPN (red) and surface γ2 (green). Changes in GPN and surface γ2 at 60 min (**E**) or 120 min (**G**) were detected by confocal microscopy. Quantitative data shown were the mean intensity (left) and cluster number (right) of GPN and surface γ2 at 60 min (**F**) and 120 min (**H**). Ctrl: 35 neurons for 60 min; 24 neurons for 120 min; LatA: 34 neurons for 60 min; 20 neurons for 120 min; Noco: 34 neurons for 60 min; 24 neurons for 120 min, from 3 independent cultures. Statistical test: one-way ANOVA by Tukey post hoc test. Scale bar, 5 µm. (**I**) Levels of synaptic proteins normalized to control neurons following LatA or Noco treatment in (**A,C,E,G**). (**J–K**) The colocalization between surface γ2 and GPN signals in dendrites in (**E**) and (**G**). Statistical test: one-way ANOVA by Tukey post hoc test. All data are scatterplots with means ± S.E.M.

The online version of this article includes the following video, source data, and figure supplement(s) for figure 6:

**Source data 1.** Original file for analysis displayed in *Figure 6B, D, F, H, I and J & K*.

**Figure supplement 1.** p140Cap promotes gephyrin clustering via endophilin A1-mediated pathway.

**Figure supplement 1—source data 1.** Original file for analysis displayed in *Figure 6—figure supplement 1C and E*.

**Figure 6—video 1.** The spatial relationship between gephyrin and LifeAct.

https://elifesciences.org/articles/102792/figures#fig6video1

---

loss-of-function mutants could rescue the GABA_AR γ2 clustering phenotype in EndoA1 KO neurons (*Figure 7C and D*). Collectively, these data indicate that both plasma membrane association and actin polymerization-promoting activities are required for endophilin A1 to promote the assembly of the inhibitory postsynaptic machinery.

Next, we determined the mechanistic role of endophilin A1 in vivo by rescuing the epilepsy susceptibility caused by EndoA1 KO. Indeed, when AAVs expressing the Cre recombinase and endophilin A1 were co-injected into the hippocampal CA1 region of *Sh3gl2^{fl/fl}* mice, only the wild-type but not the membrane-binding or p140Cap-binding deficient mutant of endophilin A1 rescued the early onset and severity of PTZ-induced seizures (*Figure 7E and F* and *Figure 7—video 1*). Consistently, electrophysiological recording in acute brain slices indicated that only wild-type endophilin A1 but not the mutants restored the amplitude of eIPSC and E/I balance in EndoA1 KO neurons (*Figure 7G–K*). All together, these data indicate that endophilin A1 contributes to the structure and function of inhibitory synapses by promoting the submembrane assembly of iPSD and synaptic localization of the GABA_ARs, thereby maintaining the balance of excitatory and inhibitory activities in neural circuits (*Figure 8*).

## Discussion

Although collybistin is implicated in plasma membrane recruitment and clustering of gephyrin and GABA_AR α2 (*Saiepour et al., 2010*), studies of collybistin KO mice showed that the GABA_AR α3 subunit promotes gephyrin-mediated formation of inhibitory synapses in the absence of collybistin (*Wagner et al., 2021*), raising the possibility that formation of a subset of GABAergic synapses requires gephyrin-interacting membrane-associated proteins other than collybistin. In the present study, we identify endophilin A1 as a novel binding partner for gephyrin and demonstrate that it promotes the assembly of inhibitory synapses and clustering of GABA_ARs at the iPSD. Loss of endophilin A1 activity causes impairment of inhibitory synaptic transmission in neuronal circuits and increases susceptibility to seizures. In the central nervous system, inhibitory interneurons are highly diverse in morphologies, molecular profiles, and physiological functions. Presynaptic terminals from different types of interneurons target distinct subcellular compartments in postsynaptic pyramidal neurons where GABA_A receptors of specific subunit compositions are located. Parvalbumin-expressing (PV+) interneurons usually contact the soma and proximal dendrites, whereas somatostatin-expressing (SST+) interneurons mainly innervate the dendrites of pyramidal cells (*Klausberger and Somogyi, 2008*). Although different inhibitory synapses localized in different subcellular compartments of postsynaptic neurons have distinct properties and express a diversity of synaptic proteins, gephyrin remains a core postsynaptic component (*Favuzzi et al., 2019*). Our study revealed that endophilin A1 promotes the clustering of gephyrin in all subcellular compartments of pyramidal neurons, indicating that endophilin A1 functions in the organization of inhibitory postsynapses innervated by various presynaptic interneurons.

Among the endophilin A proteins, endophilin A1 is the most abundant in the brain (*Milosevic et al., 2011*). Although they have redundant roles in physiological or pathological functions (*Milosevic*

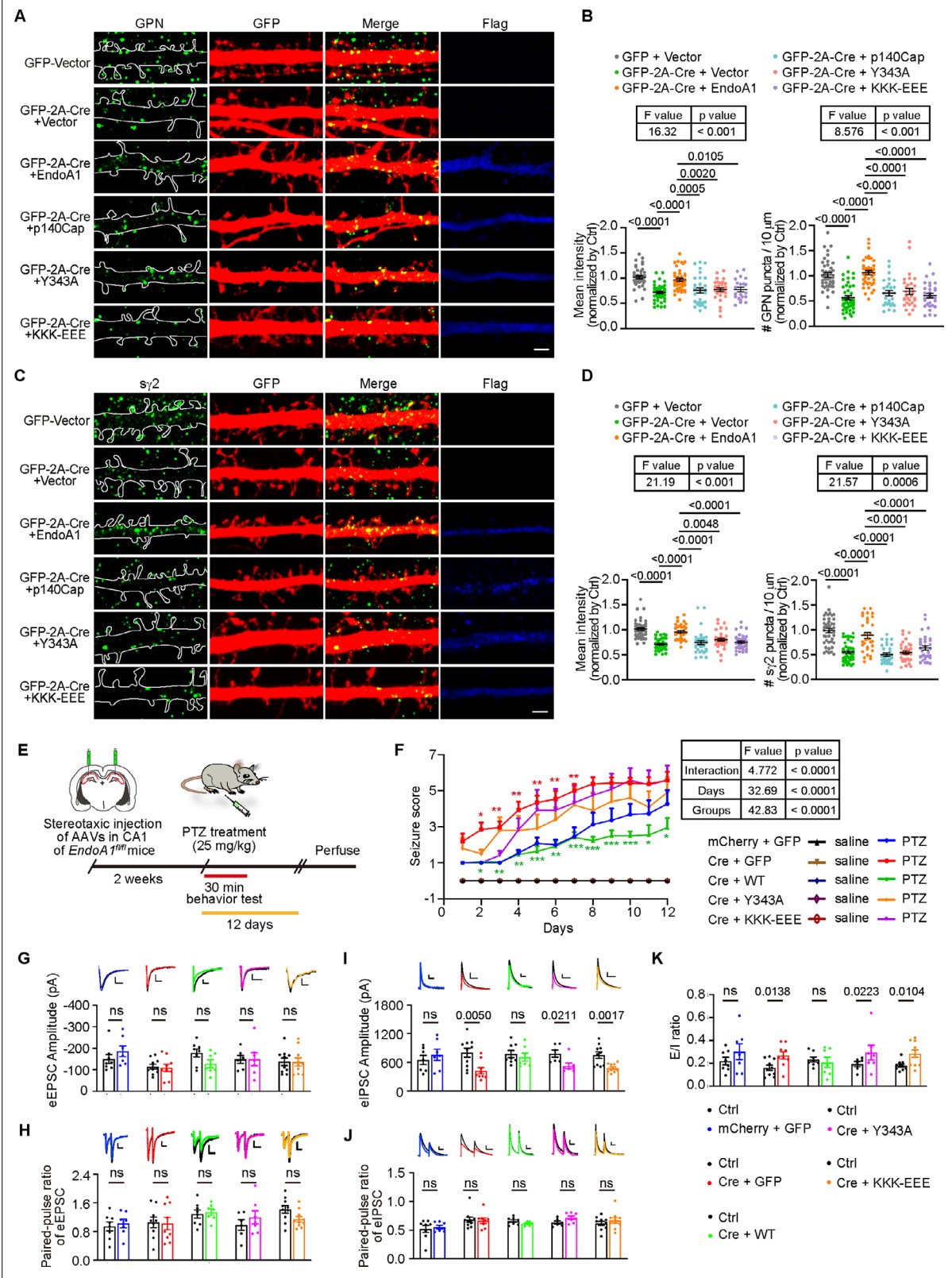

**Figure 7.** Both the membrane-binding and p140Cap-binding capacities of endophilin A1 are required for iPSD assembly and E/I balance. (**A**) Cultured hippocampal neurons from *Sh3gl2^fl/fl* mice were co-transfected with constructs expressing GFP and Flag vector or GFP-2A-Cre and Flag vector, Flag-EndoA1 wild-type (WT), mutants (KKK-EEE or Y343A) or p140Cap, and analyzed on DIV16 by immunofluorescence staining for GPN (green) and GFP (red). Scale bar, 2 μm. (**B**) Quantification of GPN mean intensity (left) and cluster number (right) in (**A**). 35 neurons for GFP vector; 43 neurons for Cre +

*Figure 7 continued on next page*

*Figure 7 continued*

vector; 32 neurons for Cre + WT; 33 neurons for Cre + KKK-EEE; 36 neurons for Cre + Y343A; 33 neurons for Cre +p140Cap, from 3 independent cultures. Statistical test: one-way ANOVA by Tukey post hoc test. (**C**) Same as (**A**) except that neurons were immunostained for surface γ2 on DIV16. Scale bar, 2 µm. (**D**) Quantification of surface γ2 mean intensity (left) and number (right) in (**C**). 44 neurons for GFP +vector; 42 neurons for Cre + vector; 32 neurons for Cre + WT; 35 neurons for Cre + KKK-EEE; 35 neurons for Cre + Y343A; 32 neurons for Cre + p140Cap, from 3 independent cultures. Statistical test: one-way ANOVA by Tukey post hoc test. (**E**) Experimental scheme for seizure scoring of *Sh3gl2fl/fl* mice stereotactically co-injected with AAVs carrying mCherry and Cre, and EGFP and EndoA1 or its mutants (KKK-EEE and Y343A) in the hippocampal CA1 region. Mice were intraperitoneally administered PTZ for 12 days after AAV injections, then seizures were scored. (**F**) PTZ-induced seizures in mice injected with the indicated AAVs were scored for 30 min. Ctrl, 7 mice; Cre, 7 mice; Cre + WT, 7 mice; Cre + KKK-EEE, 6 mice; Cre + Y343A, 7 mice. Statistical test: repeated measures ANOVA followed by Tukey post-hoc test at each time point. * $p < 0.05$, ** $p < 0.01$, *** $p < 0.001$, red color is for Cre *vs* Ctrl, green color is for Cre + WT *vs* Cre. All data are means ± S.E.M. (**G–K**) AAVs expressing Cre (AAV-mCherry-2A-Cre) and GFP (AAV-EGFP) or Cre and EndoA1 WT or mutant (AAV-EGFP-2A-EndoA1, AAV-EGFP-2A-EndoA1 KKK-EEE, AAV-EGFP-2A-EndoA1 Y343A) were stereotaxically injected into the CA1 regions of *Sh3gl2fl/fl* mouse brain at P0. Acute hippocampal slices were prepared on P14-P21 for dual recording of evoked excitatory postsynaptic currents (eEPSCs) and inhibitory postsynaptic currents (eIPSCs). Shown are representative traces and pairwise comparisons of noninfected (Ctrl) and infected neurons in the same slice. Scale bar = 50/50, 50/50, 50/50, 50/50, 40/20 (pA/ms) in (**G**); Scale bar = 100/100, 100/400, 100/100, 100/200, 300/100 (pA/ms) in (**H**); Scale bar = 50/50, 50/50, 50/50, 50/50, 50/50 (pA/ms) in (**I**); Scale bar = 100/100, 200/100, 100/100, 100/200, 100/200 (pA/ms) in (**J**). Statistical test: unpaired two-tailed Student's t-test. ns, non-significant; All data are scatterplots with means ± S.E.M. All number of recordings were from more than seven pyramidal neurons of three independent animals.

The online version of this article includes the following video and source data for figure 7:

**Source data 1.** Original file for analysis displayed in *Figure 7B*.

**Source data 2.** Original file for analysis displayed in *Figure 7D*.

**Source data 3.** Original file for analysis displayed in *Figure 7F & G–K*.

**Figure 7—video 1.** PTZ kindling behaviors of *Sh3gl2fl/fl* mice injected with AAVs expressing Cre and EndoA1 WT or mutants.
https://elifesciences.org/articles/102792/figures#fig7video1

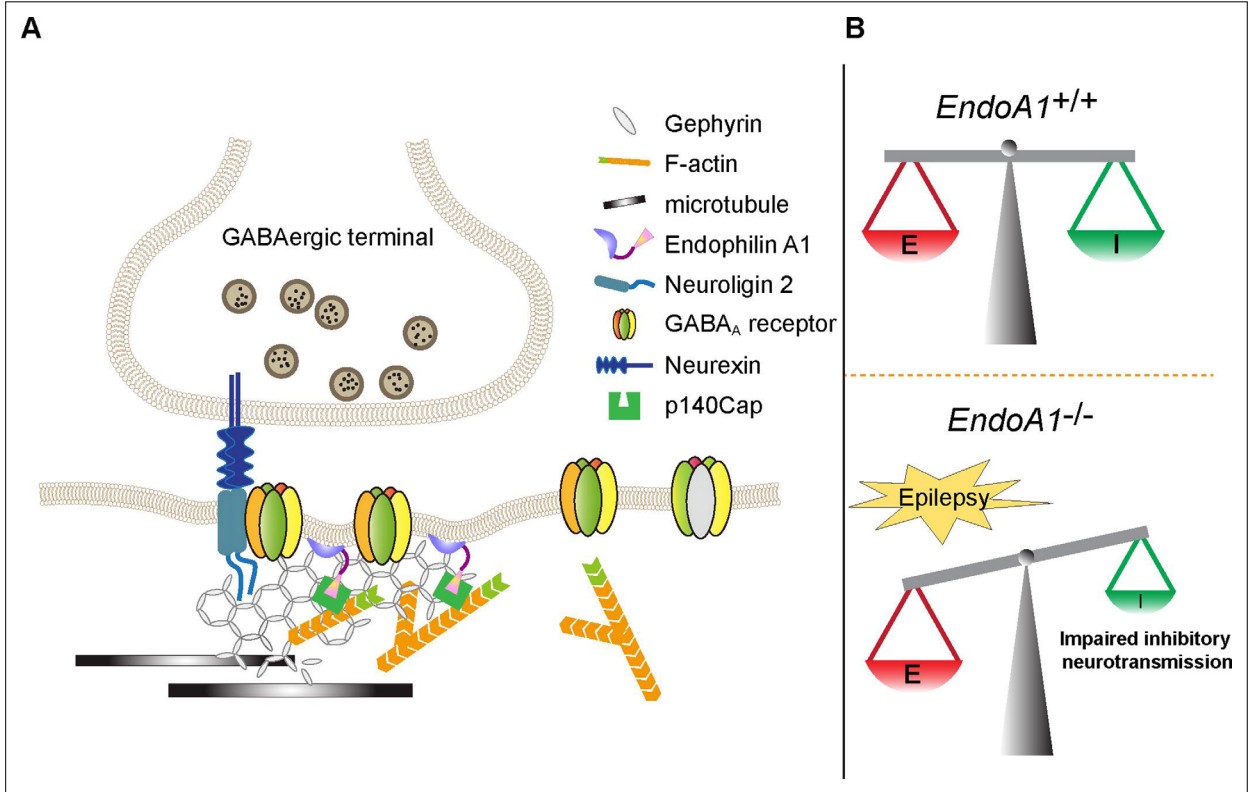

**Figure 8.** Model for the role of endophilin A1 at the inhibitory postsynapses and E/I balance. (**A**) Endophilin A1 directly interacts with the inhibitory postsynaptic scaffold protein gephyrin to promote organization of iPSD and synaptic recruitment/stabilization of the GABA_AS via its plasma membrane association and actin polymerization-promoting activities. (**B**) Loss of endophilin A1 impairs inhibitory synaptic neurotransmission and disrupts the balance of excitatory/inhibitory activities in the neural circuits, leading to enhanced epileptic seizure susceptibility.

*et al., 2011*), endophilin A1, but not A2 or A3, promotes spine morphogenesis and excitatory post-synaptic organization (*Yang et al., 2018*; *Yang et al., 2015*; *Milosevic et al., 2011*). Here, we uncover a previously unanticipated role for endophilin A1 at GABAergic synapses. By deleting endophilin As in specific neuron subtypes, we found that endophilin A1 regulates GABAergic postsynaptic compartments. Endophilin A1, rather than A2 or A3, interacts with gephyrin and modulates the inhibitory postsynaptic machinery. Although endophilin A1 might function redundantly with other endophilin As in synaptic vesicle recycling at the presynaptic site, it participates in the organization or reorganization of postsynaptic structures through specific binding to postsynaptic proteins, a function that cannot be fully replaced by other endophilin As at both excitatory and inhibitory postsynapses.

The membrane lipid-binding protein collybistin interacts with the trans-synaptic cell adhesion molecule NL-2 to recruit gephyrin and GABA$_A$R α2 to the postsynaptic membrane (*Kalscheuer et al., 2009*; *Kins et al., 2000*; *Poulopoulos et al., 2009*; *Saiepour et al., 2010*). Adamtsl3 secreted from presynapses binds to the transmembrane protein DCC (Deleted in Colorectal Cancer) and controls its synaptic localization, which is required for the maintenance of GABAergic synapses in the adult brain (*Cramer et al., 2023*). Anchoring of the α5 subunit of GABA$_A$R to the actin cytoskeleton through radixin, an actin-binding protein, facilitates its clustering (*Loebrich et al., 2006*). These findings underscore the necessity for investigation of mechanism(s) regulating the plasma membrane-iPSD-actin cytoskeleton interaction. As membrane-binding and actin polymerization-promoting activities of endophilin A1 are both required for its function in enhancing iPSD formation and γ2–containing GABA$_A$R clustering to iPSD, we propose that membrane-bound endophilin A1 promotes postsynaptic assembly by coordinating the plasma membrane tethering of the postsynaptic protein complex and its stabilization with the actin cytomatrix (*Figure 8*). In contrast, clustering of β2/3 subunit-containing GABA$_A$Rs is independent of the actin or microtubule cytoskeleton in hippocampal neurons (*Allison et al., 2000*; *Kasaragod et al., 2022*). Given that synaptic and extrasynaptic GABA$_A$Rs are composed of distinct subunits (*Kasaragod et al., 2022*; *Koh et al., 2023*), these findings indicate that the actin cytoskeleton plays distinct roles in the clustering and possibly dynamic changes in the subunit composition and subcompartmental localization of different types of GABA$_A$Rs. Of note, the subcellular distribution of gephyrin, which has a tubulin-binding motif, is not affected by depolymerization of F-actin or microtubule (*Allison et al., 2000*). Overall, the discrepancy in data about the involvement of the actin cytoskeleton in inhibitory synapses may be due to differences in drug concentration, treatment time, receptor subtypes, and neuronal cell types (*Mody and Pearce, 2004*; *Sexton et al., 2021*). In the present study, the impaired clustering of gephyrin and GABA$_A$ γ2 by F-actin depolymerization underscores the essential role of F-actin in the assembly and stabilization of the inhibitory postsynaptic machinery. Membrane-bound endophilin A1 promotes F-actin polymerization beneath the plasma membrane through its interaction with p140Cap, an F-actin regulatory protein, thereby facilitating and/or stabilizing the clustering of gephyrin and γ2-containing GABA$_A$ receptors at postsynapses.

The number of synaptic GABA$_A$ receptors is subject to alterations not only during development and neuronal plasticity but also during pathological processes such as epilepsy (*Nusser et al., 1998*). Synaptic GABA$_A$ receptors are recruited directly from extrasynaptic pools following exocytosis, and their endocytosis occurs exclusively at extrasynaptic sites (*Bogdanov et al., 2006*). Therefore, lateral diffusion of GABA$_A$Rs between synaptic and extrasynaptic sites and their iPSD anchoring are very important for their synaptic distribution (*Thomas et al., 2005*). In good agreement with the finding that GABA$_A$Rs stabilization at synapses depends on gephyrin clustering (*Jacob et al., 2005*), we found that decreases in gephyrin clustering preceded the reduction in surface GABA$_A$Rs following F-actin depolymerization (*Figure 6*). As the core scaffold protein in iPSD, gephyrin is highly enriched at the inhibitory postsynaptic sites and drives the formation of iPSD sheets via phase separation (*Bai et al., 2021*). As binding of gephyrin by its partner proteins inhibits the iPSD condensate formation via autoinhibition of the multivalent interactions between its E domain and GABA$_A$Rs (*Bai et al., 2021*; *Hoffmann and Milovanovic, 2021*; *Liu et al., 2020*), it is conceivable that endophilin A1 binding not only tethers the scaffold protein to the plasma membrane, but also releases it from the autoinhibitory conformation to facilitate the phase separation of the gephyrin-receptor complexes. Whether the activity of endophilin A1 is regulated by neuronal activity to participate in the structural and functional remodeling of iPSD awaits further investigation.

At the presynaptic site of excitatory synapses, endophilin A1 functions in synaptic vesicle recycling by recruiting the clathrin-uncoating factor synaptojanin (*Milosevic et al., 2011*; *Schuske et al.,*

*2003*; *Verstreken et al., 2003*). At the excitatory postsynapses, plasma membrane-associated endophilin A1 facilitates the morphogenesis, stabilization, maturation, and plasticity of dendritic spines by promoting actin polymerization via its interaction with p140Cap (*Yang et al., 2021*; *Yang et al., 2018*; *Yang et al., 2015*). In this study, we demonstrate that endophilin A1 functions upstream of p140Cap in not only excitatory but also inhibitory postsynapses of excitatory neurons. Intriguingly, recent studies reported that p140Cap is present in the inhibitory presynaptic compartment and regulates GABAergic synaptogenesis and neurotransmission (*Russo et al., 2019*), raising the possibility that endophilin A1 and p140Cap also coordinate to regulate the organization and activity of GABAergic synapses at the inhibitory presynaptic site. Remodeling of inhibitory synapses and dendritic spines is spatially clustered by sensory inputs in the central nervous system (*Chen et al., 2012*). Notably, the turnover of inhibitory synapses outpaces that of excitatory synapses (*Chen et al., 2012*). Therefore, it is conceivable that deficits in inhibitory synapses are much more severe in EndoA1 KO mice, leading to perturbation of the E/I balance and increased epileptic vulnerability.

## Materials and methods
### Animals
Generation of *Sh3gl2^{fl/fl}* and *Sh3gl2^{-/-}* (KO first) mice on the C57BL/6J background was as previously described (*Yang et al., 2018*). The targeting vector for *Sh3gl2* was obtained from European Mouse Mutant Cell Repository (EuMMCR, PRPGS00060_A_A02). The *Sh3gl2^{-/-}* and *Sh3gl2^{fl/fl}* C57BL/6J mice were generated at Nanjing Biomedical Institution of Nanjing University. *Sh3gl2^{+/+}* and *Sh3gl2^{-/-}* mice were bred as littermates from heterozygous breeders. Genotyping of mouse lines was performed by genomic PCR of tail prep DNA from offspring with the following primer pairs: loxPF/loxPR: 5'-CAAG GACTCCCAGAGACCTAGCATC-3' and 5'-GAGATGGCGCAACGCAATTAAT-3' (A PCR product of 375 base pairs in EndoA1 KO first mice but none in wild-type mice). zptF/zptR: 5'-GTAAGCGGCTCT AGCGCATGTTCT-3' and 5'-GCAGGGGCATGTAGGTGGCTCAAC-3' (A PCR product of 466 base pairs in wild-type mice, none in *Sh3gl2^{-/-}* mice, and of 627 base pairs in *Sh3gl2^{fl/fl}* mice).

### Constructs and viruses
pCMV-Tag2B-EndoA1, pCMV-Tag2B-p140Cap, pCMV-Tag2B-EndoA1 Y343A, pCMV-Tag2B-EndoA1 KKK-EEE, pCMV-Tag2B-EndoA1 (aa172-352), pGEX-4T-1-SH3 (aa295-352), pET-28a (+)-EndoA1 and LifeAct-mCherry constructs were described previously (*Yang et al., 2021*; *Yang et al., 2015*). pAOV-CaMKIIα-mCherry-2A-3FLAG were purchased from OBiO Technology (Shanghai) Corp. Ltd. (Shanghai, China). pNICE-NL2(-) was a gift from Peter Scheiffele (Addgene plasmid # 15246) (*Chih et al., 2006*). pCR3-FLAG-Gephyrin P1 was a gift from Shiva Tyagarajan (Addgene plasmid # 68816) (*Tyagarajan et al., 2013*). pCAG_PSD95.FingR-eGFP-CCR5TC and pCAG_GPN.FingR-eGFP-CCR5TC were gifts from Don Arnold (Addgene plasmid # 46295 & #46296) (*Gross et al., 2013*). The pCMV-Tag3B-GPN construct was generated by cloning gephyrin cDNA amplified from pCR3-FLAG-gephyrin into pCMV-Tag3B (Clontech Laboratories, Inc). The pGEX-4T-1 EndoA1 ΔBAR (Δaa6-242) and pGEX-4T-1 EndoA1 ΔSH3 (Δaa295-346) constructs were generated by cloning fragments amplified from pCMV-Tag2B-EndoA1 into pGEX-4T-1. The pGEX-4T-1 GPN, pGEX-4T-1 GPN G (aa15-169), pGEX-4T-1 GPN C (aa170-326) and pGEX-4T-1 GPN E (aa327-733) constructs were generated by PCR amplification from full-length constructs and subcloned into pGEX-4T-1. All constructs were verified by DNA sequencing. Viral particles of adeno-associated virus (AAV) carrying mCherry, EGFP, the Cre recombinase, EndoA1 WT, or mutants were purchased from OBiO Technology (Shanghai) Corp. Ltd. (Shanghai, China). Viral particles of rAAV-EF1a-PSD95.FingR-eGFP-CCR5TC and rAAV-EF1a-mRuby2-Gephyrin.FingR-IL2RGTC (*Bensussen et al., 2020*) were purchased from brain case Corp. Ltd. (Shenzhen, China).

### Antibodies
The following antibodies were obtained from commercial sources: rabbit anti-endophilin A1 (159 002, Synaptic Systems GmbH, Germany), mouse anti-Gephyrin (147 111, Synaptic Systems), guinea pig anti-gephyrin (147 318, Synaptic Systems), mouse anti-neuroligin 2 (129 511, Synaptic Systems), mouse anti-GluA1 (MAB2263, Millipore), rabbit anti-GABA$_A$R α1 (06-868, Millipore Sigma), rabbit anti-GABA$_A$R γ2 (224 003, Synaptic Systems), mouse anti-SYP (D-4) (sc-17750, Santa Cruz), rabbit anti-FLAG (F3165, Sigma-Aldrich), rabbit and mouse anti-RFP which recognizes DsRed, mCherry, and

mRuby2 (PM005 and M165-3, Medical & Biological Laboratories, Naka-ku Nagoya, Japan); mouse anti-α-tubulin (T9026, Sigma-Aldrich), mouse anti-PSD95 (610495, BD Biosciences, San Diego, CA) and mouse anti-GST (M071-3, MBL) for western blotting (WB) and rabbit anti-VGAT (131 004, Synaptic Systems), mouse anti-PSD95 (75-028, NeuroMab, Davis, CA), guinea pig anti-Ankyrin G (386 004, Synaptic Systems), guinea pig anti-GABA$_A$R γ2 (224 004, Synaptic Systems) and mouse anti-Bassoon (sc-58509, Santa Cruz) for immunofluorescence staining; Rabbit anti-p140Cap was described previously (*Yang et al., 2015*). Alexa Fluor dye-conjugated secondary antibodies for immunofluorescence staining were from Molecular Probes (Invitrogen, Carlsbad, CA, USA).

## Pentylenetetrazole (PTZ)-kindling and seizure behavior scoring

To evaluate seizure susceptibility, 8- to 10-week-old male and female *Sh3gl2$^{+/+}$* or *Sh3gl2$^{-/-}$* littermates or *Sh3gl2$^{fl/fl}$* littermates were intraperitoneally administered with PTZ (25 mg/kg; K0250, Sigma-Aldrich) or saline (control) for 8-12 days, and the resulting seizure behaviors were video-recorded for the next 30 min. Seizure susceptibility was measured blindly by rating seizures on a scale of 0 to 7 as follows (*Van Erum et al., 2019*): no abnormal behavior (0), reduced motility and prostate position (1), facial jerking (*Schulz et al., 2021*), neck ierks (*Huo et al., 2009*), clonic seizure (sitting) (4), clonic, tonic-clonic seizure (lying on belly) (5), clonic, tonic-clonic seizure (lying on side) and wild jumping (6), tonic extension, possibly leading to respiratory arrest and death (7).

## Stereotaxic injection

For virus injection, 8- to 9-week-old naive male and female littermates were anesthetized and stereotactically injected with viral particles in the hippocampal CA1 region as described (*Yang et al., 2018*). Briefly, mice were weighed and anesthetized with isoflurane (2% 2,2,2-tribromoethanol, Sigma), and secured in a stereotaxic apparatus. The scalp of the mouse was cut along the midline between the ears after sterilized with iodophors and 75% (vol/vol) alcohol. Holes for inserting the needle were drilled in the bilateral skull. Viral particles carrying pAOV-CaMKIIα-MCS-mCherry-3FLAG or pAOV-CaMKIIα-mCherry-2A-Cre and pAAV-CaMKIIα-EGFP-2A-MCS-3FLAG or pAAV-CaMKIIα-EGFP-2A-MCS-3FLAG-EndoA1 WT or mutants (1.2 µL, 2.0× 10$^{12}$ viral genomes/mL) (OBiO Technology Corp., Ltd) were injected bilaterally into the hippocampal CA1 regions using the following coordinates relative to bregma: 2.0 mm posterior, 1.8 mm lateral, and 1.4 mm ventral at a rate of 0.125 µL/min using a microinjection system (World Precision Instruments). The needle was kept in place for 2 min before withdrawal, and the skin was sutured with surgical line. The virus-injected mice were tested for behavior two weeks later.

## Immunofluorescence staining of brain sections

Male and female littermates (P21 or P60) were anesthetized and immediately perfused, brains were dissected out, fixed overnight in 4% (vol/vol) paraformaldehyde, incubated overnight with 30% sucrose in phosphate-buffered saline (PBS), and then sliced into 30 µm-thick coronal sections using a cryotome (Model CM-1950; Leica Biosystems). For immunostaining of brain sections, floating 30 µm-thick slices were rinsed with PBS and permeabilized in 0.4% Triton X-100 in 0.01 M PBS for 30 min. Cryosections were blocked with 2% BSA in PBS containing 0.4% Triton X-100 for 2 hr at room temperature (RT), then incubated with primary antibodies overnight at 4°C. Secondary antibodies conjugated with Alexa Fluor 488 or 555 were used for detection. Sections were then incubated with DAPI (Roche, Grenzach-Wyhlen, Germany) for nuclear staining for 3 min at RT. Following rinsing, cyrosections were mounted on gelatin-coated slides and covered with coverslip with mounting medium. Confocal images were collected using the Spectral Imaging Confocal Microscope Digital Eclipse A1Si (Nikon, Tokyo, Japan) with a 40× Plan Fluo (NA 1.30) oil objective (*Yang et al., 2018*).

## Cell culture, transfection, and drug treatment

Primary hippocampal neurons were cultured as previously described (*Yang et al., 2021*). Briefly, hippocampi of female or male pups (P0) were rapidly dissected under sterile conditions, kept in cold DMEM (4°C) with high glucose. Tissue was then digested with 0.25% trypsin solution at 37°C for 18-20 min, triturated in DMEM supplemented with 10% F12 (11765054, Gibco) and 10% FBS (Gibco, Carlsbad, CA, USA). Then the tissue was mechanically dissociated into single cells by gentle pipetting up and down. Neurons were plated at a density of ~3.0-3.5 × 10$^5$ cells/well in poly-D-lysine (P6407,

Sigma-Aldrich) -coated six-well plate and at a density of 2.5-3.0 × $10^4$ cells/well in 24-well plate. Four hours later, media was changed to Neurobasal A (NB-A) growth medium supplemented with 2% B27 supplement (17504-044, Gibco), 1% GlutaMAX (35050-061, Gibco), and 33 mM glucose. Culture media were changed by half volume with Neurobasal maintenance media once a week. The cultures were maintained at 37°C with 5% $CO_2$. Experiments were performed at DIV14-16 unless otherwise stated. For neuronal transfection, hippocampal neurons (DIV11) were transfected using Lipofectamine-LTX (15338100, Invitrogen, ThermoFisher Scientific). For 12 mm coverslips in individual wells of a 24-well plate, 0.5 µg of DNA was mixed with 0.5 µL PLUS reagent in 50 µL of NB-A and incubated for 5 min at RT, then gently mixed with 1 µL Lipofectamine LTX in 50 µL NB-A and incubated for 20 min at RT. The DNA-Lipofectamine LTX mixture was then added to the neurons in NB-A at 37°C in 5% $CO_2$ for 1 hr. Neurons were then rinsed with NB-A and incubated in the original medium at 37°C in 5% $CO_2$ for 3-4 days.

HEK293T cells for co-IP (Cat. No. 632180, Takara) were authenticated by short tandem repeat (STR) analysis (Guangzhou Cellcook Biotech Co., Ltd). Mycoplasma contamination was regularly tested by PCR and ensured that all cells were not contaminated with mycoplasma throughout the culture period. Cells were cultured in DMEM supplemented with 10% FBS. Cell transfections were performed using Lipofectamine-2000 (11668019, Invitrogen, ThermoFisher Scientific) according to the manufacturer's instructions. Cells were harvested 24-48 hr post-transfection.

For inhibitor treatment, primary neurons cultured on coverslips were pre-incubated with Latrunculin A (1 µM, 76343-93-6, Sigma-Aldrich), Nocodazole (1 µM, 31430-18-9, Sigma-Aldrich) for 60 min or 120 min.

## Immunofluorescence staining of cultured neurons

Hippocampal neurons were fixed in 4% paraformaldehyde/4% sucrose in PBS at RT for 15 min, then permeabilized in 0.4% Triton X-100 in PBS for 10 min. After blocking with 2% BSA in PBS containing 0.4% Triton X-100 for 30 min at RT, neurons were incubated with primary antibodies for 1 hr at RT or overnight at 4°C, and appropriate secondary antibodies conjugated with Alexa Fluor 488, Alexa Fluor 555, or Alexa Fluor 647 were used for detection. The primary antibody was diluted in a PBS-2% BSA solution and incubated overnight at 4°C. After washing with PBS three times, samples were incubated with Alexa Fluor secondary antibodies (1:2000, Thermo Fisher) for 1 hr at RT and mounted without bubbles.

## Biotinylation of cell surface proteins

Surface biotinylation experiments were performed as previously described (*Guo et al., 2022*). Briefly, neurons were washed with ice-cold extracellular solution (ECS) (in mM: 125 NaCl, 2 $CaCl_2$, 2.5 KCl, 5 HEPES, 33 glucose, pH, 8.0) and incubated for 30 min with 0.8 mg/ml NHS-SS-Biotin (21327, Thermo Fisher Scientific) in ECS buffer at 37°C. After incubation, cells were washed once with PBS, and the unbound biotin was quenched via two 8 min incubations with quenching buffer (20 mM glycine in PBS). Lysis was performed by mechanical scraping in ice-cold lysis buffer (50 mM Tris-Cl pH 7.4, 150 mM NaCl, 1% TritonX-100, 0.1% SDS). After rotating for 30 min at 4°C, lysates were centrifuged for 10 min at 14,000× *g* at 4°C. Balanced Pierce Streptavidin Agarose beads (25 µL, 20353, Thermo Fisher Scientific) were added to the supernatant. The mixture of lysates and beads was rotated at 4°C overnight and then washed twice with ice-cold lysis buffer. Beads were incubated with 2× SDS sample buffer for 10 min at 37°C and biotinylated proteins were analyzed by immunoblotting.

## Protein expression and purification

GST-tagged EndoA1 or gephyrin fragments was expressed in *E. coli* BL21 (DE3). Cells were grown at 37°C in LB (g/L: tryptone 10, yeast extract 5, NaCl 10) supplemented with ampicillin. Cells were induced at OD600 of ~0.6 with 0.4 mM isopropyl-D-1-thiogalactopyranoside (IPTG, A600168-0005, Bio Basic Inc) for 4 hr at 30°C or 16 hr at 16°C. Cell pellets were resuspended in PBS supplemented with 0.2% Triton X-100 and 0.1 mM PMSF. Proteins were purified with Glutathione Sepharose 4B (GE17-0756-01, Sigma-Aldrich, St. Louis, MO, USA) according to the manufacturer's instructions.

## GST-pull down, co-immunoprecipitation (IP), and immunoisolation

For GST-pull down assays, GST-fused protein EndoA1 and its fragments were expressed and purified from *E. coli*. Myc-tagged GPN constructs were expressed in HEK293T cells for 48 hr and lysates were prepared in a buffer (in mM: Tris-HCl 50, pH 7.4, NaCl 150, 1% NP-40, EDTA 2) plus protease inhibitors cocktail. Cell lysates were centrifuged at 12,000× *g* for 15 min at 4°C and the supernatant was incubated with individual GST-fused proteins conjugated with Glutathione-Sepharose beads overnight at 4°C. Beads were washed five times with lysis buffer and boiled in 2× SDS sample buffer. Bound proteins were analyzed by immunoblotting.

For coIP from brain lysates, the whole brain from 8- to 10-week-old WT and KO littermates was dissected and homogenized in the pre-chilled lysis buffer (in mM: Tris-HCl 20, pH 7.4, NaCl 150, 1% NP-40, EDTA 5) plus protease inhibitor cocktail, followed by incubation on ice for 15 min. The lysates were then centrifuged at 12,000× *g* for 15 min and the supernatant was collected and incubated with anti-EndoA1 antibody (2 µg) or control IgG overnight at 4°C, followed by incubation with 20 µL Dynabeads Protein A (ThermoFisher, 10007D for rabbit antibody) for 2 hr at 4°C. After washing the beads with the same lysis buffer four times, the bound proteins were incubated with 2× SDS sample buffer at 37°C for 10 min and subjected to SDS-PAGE and western blot with indicated antibodies.

For coIP of proteins in HEK293T cells, cells were lysed with lysis buffer (in mM: 0.5% NP-40, Tris-HCl 50, pH 7.4, NaCl 150, EDTA 5) supplemented with protease inhibitors. Lysates were then centrifuged at 16,000× *g* for 15 min at 4°C. Cell lysates were incubated with anti-Flag Affinity Gel (A2220, Sigma-Aldrich) overnight at 4°C on a roller mixer. Immunoprecipitates were washed four times in lysis buffer and boiled in 2× SDS sample buffer, then subjected to SDS-PAGE and immunoblotting.

For immunoisolation of membrane proteins, hippocampi were homogenized with lysis buffer (in mM: Tris-HCl 20, pH 7.4, HEPES 10, pH 7.4, NaCl 150, sucrose 250, EDTA 0.5, pH 8.0) supplemented with protease inhibitors and centrifuged at 800× *g* for 15 min. The supernatants were collected and subjected to high-speed centrifugation at 100,000× *g* for 1 hr (TLS-55 rotor, OptimaTMMAX Ultracentrifuge; Beckman Coulter, USA). The supernatants (S100) and pellets (P100, the membrane fraction) resuspended in lysis buffer were subjected to immunoisolation with Dynabeads Protein A (100-4D, Invitrogen, Carlsbad, CA, USA) coupled with 2 µg of rabbit anti-EndoA1 or rabbit anti-GABA$_A$R α1 antibody. Bound proteins were eluted by incubation with 2× SDS loading buffer at 37°C for 10 min and subjected to SDS-PAGE and immunoblotting.

## PSD fractionation of brain lysates

PSD fractions from adult mouse brain were prepared as previously described (*Yang et al., 2015*). In brief, hippocampi were homogenized on ice with 20 strokes of a Teflon-glass homogenizer in 1 mL of HEPES-buffered sucrose (0.32 M sucrose, 4 mM HEPES, pH 7.4) containing freshly added protease inhibitors, then centrifuged at 800× *g* at 4°C to remove the pelleted nuclear fraction (P1). Supernatant (S1) was centrifuged at 10,000× *g* for 15 min to yield the crude synaptosomal pellet (P2), and the pellet was washed once in 1 mL HEPES-buffered sucrose. P2 was lysed by hypoosmotic shock in 900 µL ice-cold 4 mM HEPES, pH 7.4 plus protease inhibitors, homogenized by pipetting and rotating for 30 min at 4°C. The lysate was centrifuged at 25,000× *g* for 20 min to yield supernatant (S3, crude synaptic vesicle fraction) and pellet (P3, lysed synaptosomal membrane fraction). To prepare the PSD fraction, P3 was resuspended in 900 µL of ice-cold 50 mM HEPES, pH 7.4, 2 mM EDTA, plus protease inhibitors and 0.5% Triton X-100, rotated for 15 min at 4°C and centrifuged at 32,000× *g* for 20 min to obtain the PSD pellet. PSD pellets were resuspended in 50 µL ice-cold 50 mM HEPES, pH 7.4, 2 mM EDTA plus protease inhibitors. Proteins were subjected to SDS-PAGE and immunoblotting.

## Protein identification with mass spectrometry and pathway analysis

For identification of EndoA1-interacting proteins, mouse brain was homogenized with lysis buffer (in mM: Tris-HCl 20, pH 7.4, HEPES 10, pH 7.4, NaCl 150, EDTA 1, 1% NP-40) supplemented with protease inhibitors. Lysates were centrifuged at 12,000× *g* for 15 min. The supernatants were incubated with rabbit anti-EndoA1 antibody overnight at 4°C. Beads were rinsed with lysis buffer five times and bound proteins were resolved by SDS-PAGE. Proteins in the gels were reduced with 10 mM DTT at 37°C for 1 hr, then alkylated with 25 mM iodoacetamide at RT for 1 h in the dark before digestion with trypsin (T1426, Sigma-Aldrich; enzyme-to-substrate ratio 1:50) in 25 mM ammonium bicarbonate at 37°C overnight. Tryptic peptides were extracted from gel by sonication with a buffer containing 5%

trifluoroacetic acid and 50% acetonitrile. The liquid was dried by SpeedVac, and peptides were resolubilized in 0.1% formic acid and filtered with 0.45 µm centrifugal filters before analysis with a TripleTOF 5600 mass spectrometer (AB SCIEX, Canada) coupled to an Eksigent nanoLC. Proteins were identified by searching the MS/MS spectra against the *Mus musculus* SwissProt database using the ProteinPilot 4.2 software. Carbamidomethylation of cysteine was set as the fixed modification. Trypsin was specified as the proteolytic enzyme with a maximum of 2 missed cleavages. Mass tolerance was set to 0.05 Da and the false discovery rates for both proteins and peptides were set at 1%. For enrichment pathway analysis: Only proteins with intensity to IgG ratio of > 1.5 were considered in the analysis. These genes were analyzed by Gene Ontology (GO) Enrichment Analysis (http://bioinfo.org/kobas).

## Electrophysiology

To knockout EndoA1 in the brain, *Sh3gl2*$^{fl/fl}$ mice were injected with high-titer AAV stock carrying pAOV-CaMKIIα-mCherry-2A-Cre (AAV-mCherry-2A-Cre) (1~5 × 10$^{13}$ IU/ml) in the hippocampal CA1 region within 24 hr after birth. Newborn *Sh3gl2*$^{fl/fl}$ littermates (male or female) were anesthetized on ice for 4-5 min and then mounted in a custom ceramic mold to align the head along the X and Y axes, with lambda set as (X, Y) = (0, 0). The zero point of the Z axis was the position at which the injecting needle penetrated the skin. Approximately 10 nL of the viral solution was injected at each of the seven sites (coordinates in mm, [X, Y, Z] = [1.2, 1.2, 1.4/1.0/0.6] and [1.5, 1.0, 1.7/1.3/0.9/0.5]) targeting the hippocampus in each cerebral hemisphere using a microsyringe (Sutter Instrument, Novato, CA, USA) and beveled glass injection pipette (*Yang et al., 2018*). For the rescue experiment, P0 *Sh3gl-2*$^{fl/fl}$ mice were injected with a mixture of two high-titer AAV stocks: AAV-EGFP and AAV-mCherry or AAV-mCherry-2A-Cre; AAV-mCherry-2A-Cre and AAV-EGFP-2A-FLAG-EndoA1, AAV-EGFP-2A-FLAG-EndoA1 Y343A or AAV-EGFP-2A-FLAG-EndoA1 KKK-EEE. Approximately 20 nL of the viral solution was injected at each of the seven sites. The injected pups were returned to the home cage and used for recording 2 weeks later.

To prepare acute brain slices for electrophysiological analyses, mice were anesthetized with tribromoethanol (330-390 mg/kg) (T48402, Sigma-Aldrich, St. Louis, MO, USA) and decapitated. Whole brains were removed promptly and placed in ice-cold oxygenated (95% O$_2$/5% CO$_2$) low-Ca$^{2+}$/high-Mg$^{2+}$ high-sucrose cutting solution (in mM: KCl 2.5, NaHCO$_3$ 25, NaH$_2$PO$_4$ 1.25, glucose 10, sucrose 210, Na-Ascorbate 1.3, CaCl$_2$ 0.5, MgSO$_4$ 7). Coronal 300 mm slices were obtained using a vibrating tissue slicer (Leica VT1200S; Leica Biosystems, Wetzlar, Deutschland). Slices were immediately transferred to standard artificial cerebrospinal fluid (ACSF; in mM: NaCl 119, KCl 2.5, MgSO$_4$ 1.3, CaCl$_2$ 2.5, glucose 11, NaHCO$_3$ 26.2, NaH$_2$PO$_4$ 1, pH 7.3-7.4) at 33°C and continuously bubbled with 95% O$_2$/5% CO$_2$. After 30 min of incubation, slices were transferred to a recording chamber with the same extracellular buffer at room temperature (RT, ~25°C).

For whole-cell patch clamp recordings, brain slices were continuously perfused with standard ACSF at RT under visual control with differential interference contrast illumination on an upright microscope (BX51WI, Olympus, Tokyo, Japan) with a 40× water-immersion objective and a digital camera. The mCherry- and/or EGFP-positive hippocampal CA1 pyramidal neurons were identified by an epifluorescence microscope (pE-300, CoolLED Ltd, Andover, UK). Whole-cell recordings were performed on an AAV-infected neuron and a neighboring uninfected neuron. Patch clamp data were obtained using a Multiclamp 700B amplifier (Molecular Devices, San Jose, CA, USA), and signals were low-pass filtered at 2 kHz and digitized at 10 kHz using an Axon Digidata 1550B analog-to-digital board (Molecular Devices, San Jose, CA, USA).

To analyze evoked excitatory postsynaptic currents (eEPSCs), evoked inhibitory postsynaptic currents (eIPSCs), and paired-pause ratio, cells were recorded with 3-7 MW borosilicate glass patch pipettes filled with the intracellular solution (in mM: CsMeSO$_3$ 135, HEPES 10, EGTA 1, QX-314 3.3, MgATP 4, NaGTP 0.3, and Na$_2$-phosphocreatine 8, pH 7.3) (*Horn and Nicoll, 2018*), following stimulation of Schaffer collaterals with a bipolar stimulation electrode placed in the stratum radiatum of the CA1 region. EPSCs were measured at a holding potential of –70 mV, and IPSCs were recorded at +10 mV. Paired-pulse ratios of EPSCs were measured at a holding potential of –70 mV by applying two electrical stimulations at a 50 ms interval and calculating the ratio of the peak amplitudes. Paired-pulse ratios of IPSCs were measured at a holding potential of +10 mV by applying two electrical stimulations at a 200 ms interval and calculating the ratio of the peak amplitudes. E/I ratios were measured by applying the same stimulation current and then calculating the ratio of the absolute values of the

peak amplitudes of EPSCs and IPSCs. The miniature EPSCs and IPSCs were measured in the ACSF bath solution containing 1 mM tetrodotoxin (TTX, MSS0002, BGB Analytik AG, BL, Switzerland) at –70 mV and +10 mV, respectively, in the voltage-clamp mode. Data were collected and analyzed using the pClamp 10.6 software (Molecular Devices, San Jose, CA, USA).

All electrophysiological recordings were carried out blindly, and all n numbers in the data figures indicate the number of recorded neurons. All chemicals were purchased from Sigma-Aldrich except otherwise noted.

## Confocal and three-dimensional structured illumination microscopy (3D-SIM) image acquisition and analysis

Confocal images were collected using the Spectral Imaging Confocal Microscope Digital Eclipse A1Si (Nikon, Tokyo, Japan) with a 100× Plan Apochromat VC (NA 1.40) oil objective (4× optical zoom). Images were z projections of images taken at 0.25-μm step intervals. The number of planes, typically 5-7, was chosen to cover the entire dendrite from top to bottom. Images were captured using a 1024×1024 pixels screen for neuronal cells. Laser power, digital gain, and offset settings were identical in each experiment. Images were processed with the NIH ImageJ software and exported into Adobe Photoshop. Only brightness and contrast were adjusted for the whole frame, and no part of a frame was enhanced or modified in any way. Control and experimental group neurons which were to be directly compared were imaged with the same acquisition parameters.

For 3D-SIM, secondary antibodies conjugated with Alexa Fluor 568 were used and images were captured as described by (*Niu et al., 2017*) on the DeltaVision OMX V4 imaging system (Applied Precision Inc, USA) with a 100× 1.4 oil objective (Olympus UPlanSApo), solid state multimode lasers (488, 593, and 642 nm) and electron-multiplying charge-coupled device (CCD) cameras (Evolve 512× 512, Photometrics, USA). Serial Z-stack sectioning was done at 125 nm intervals. The microscope is routinely calibrated with 100 nm fluorescent spheres to calculate both the lateral and axial limits of image resolution. SIM image stacks were reconstructed using softWoRx 5.0 (Applied Precision) with the following settings: pixel size 39.5 nm; channel-specific optical transfer functions; Wiener filter 0.001000; discard Negative Intensities background; drift correction with respect to first angle; custom K0 guess angles for camera positions. Pixel registration was corrected to be less than 1 pixel for all channels using 100 nm Tetraspeck beads. Three-dimensional images were reconstructed from z stacks. For clarity of display, small linear changes of brightness and contrast were performed on three-dimensional reconstructions.

Quantification of immunostaining was carried out blindly. For quantitative analysis of immunofluorescent signals in neurons, maximal projection images were created with the ImageJ software (https://imagej.net.) from 6 to 7 serial optical sections. Background was subtracted by using the 'subtract background' function and the background level was held identical for all cells within each experiment. To examine the number and fluorescence intensity of signal puncta in soma or dendrites, the EGFP- or DsRed-labeled soma or dendrites were outlined automatically by using the 'Wand Tool' function. The region of interest (ROI) in cultured neurons was defined along a segment of the dendrite according to the fluorescence signal distinguished from the background. Average values of fluorescence intensities in ROI (the total fluorescent intensity divided by the total area of a dendritic segment) were calculated by Analyze > measure. Gephyrin clusters were thresholded and confirmed visually to select appropriate clusters following a minimal size cut-off by using the 'Analyze Particles' plugin of the ImageJ software. For colocalization quantifications, NIS-Elements Software was used as follows: region of interest (ROI) of dendritic segments was manually chosen and thresholded using the 'Threshold' function to remove background signals. Pearson's correlation coefficient and Manders overlap coefficient were analyzed by using the 'Analyze Control > colocalization' function.

For spatial correlation/colocalization analysis in super-resolution images, we used the method described in the reference (*McCall, 2024a*). Briefly, channels in 3D images were split and followed by Gaussian blur performed on both GPN and Bassoon channels. Next, masks of all possible localizations for the signal in GPN and Bassoon image channel were obtained. Then, the input images of the three channels were deconvolved using 10 iterations of the Richardson-Lucy algorithm in Deconvolution Lab 2 (version 2.1.2), with theoretical point spread functions (PSFs) generated by the PSF Generator plugin. PSFs were generated using the Born & Wolf model, with parameters matching the acquisition parameters: 1.518 immersion refractive index, 1.4 NA, 40 nm pixel size and 125 nm z-step, 512 × 512

× 14 output diameters, and 520 nm, 592 nm, and 670 nm wavelengths. Finally, select the two images (GPN and EndoA1 or Bassoon and EndoA1) and the mask images, then perform a cross-correlation analysis (CCC) between the two images using the CCC plugin for ImageJ software(https://github.com/andmccall/Colocalization_by_Cross_Correlation; *McCall, 2024b*), which is publicly available under GPL-3.0 license.

## Statistical analysis

For all biochemical, cell biological, and electrophysiological recordings, at least three independent experiments were performed (independent cultures, transfections, or different mice). Statistical analysis was performed in GraphPad Prism 9.0 (GraphPad Software, La Jolla, CA). Data sets were tested for normality and direct comparisons between two groups were made using two-tailed Student's t test (t test, for normally distributed data) as indicated. To evaluate statistical significance of three or more groups of samples, one-way ANOVA analysis with a Tukey test was used or repeated measures ANOVA analysis with a Tukey test was used in behavior assays. Statistical parameters are reported in the figures and the corresponding legends. The statistical significance was defined as $* \ p < 0.05$, $** \ p < 0.01$, $*** \ p < 0.001$, respectively. All data are presented as scatterplots with mean ± S.E.M.

## Acknowledgements

We thank Drs Shiva Tyagarajan (Addgene #68816), Peter Scheiffele (Addgene # 15246) and Don Arnold (Addgene # 46295 & #46296) for constructs. This work was supported by funding from the National Natural Science Foundation of China (NSFC 32271005 and 32070785 to Y Yang, 31921002, 32270730 and 91954126 to J-J Liu) and the State Key Laboratory of Molecular Developmental Biology (2022-MDB-KF-13).

## Additional information

### Funding

| Funder | Grant reference number | Author |
| --- | --- | --- |
| National Natural Science Foundation of China | 32271005 | Yanrui Yang |
| National Natural Science Foundation of China | 32070785 | Yanrui Yang |
| National Natural Science Foundation of China | 31921002 | Jia-Jia Liu |
| National Natural Science Foundation of China | 32270730 | Jia-Jia Liu |
| National Natural Science Foundation of China | 91954126 | Jia-Jia Liu |
| State Key Laboratory of Molecular Developmental Biology | 2022-MDB-KF-13 | Jia-Jia Liu |

The funders had no role in study design, data collection and interpretation, or the decision to submit the work for publication.

### Author contributions

Xue Chen, Data curation, Formal analysis, Writing – original draft; Deng Pan, Data curation, Formal analysis; Jia-Jia Liu, Supervision, Funding acquisition, Project administration, Writing – review and editing; Yanrui Yang, Conceptualization, Supervision, Funding acquisition, Writing – original draft, Project administration, Writing – review and editing

### Author ORCIDs

Xue Chen ⓘ https://orcid.org/0009-0006-9246-2182
Deng Pan ⓘ https://orcid.org/0009-0001-6540-7587

Jia-Jia Liu https://orcid.org/0000-0002-6099-1059
Yanrui Yang https://orcid.org/0000-0001-5194-1239

### Ethics

All animal experiments were approved by and performed in accordance with the guidelines of the Animal Care and Use Committee of Institute of Genetics and Developmental Biology, Chinese Academy of Sciences (Approval code: AP2022008). All animals were housed in standard mouse cages at 22-24°C on a 12 h light/dark cycle with access to food and water freely.

Reviewer #1 (Public review): https://doi.org/10.7554/eLife.102792.4.sa1
Reviewer #2 (Public review): https://doi.org/10.7554/eLife.102792.4.sa2
Reviewer #3 (Public review): https://doi.org/10.7554/eLife.102792.4.sa3
Author response https://doi.org/10.7554/eLife.102792.4.sa4

## Additional files

### Supplementary files

Supplementary file 1. Proteins identified by mass spec analysis of endophilin A1 co-immunoprecipitates and their abundance. Related to *Figure 4*.

MDAR checklist

Source data 1. Primers used in the current research.

### Data availability

All data generated or analysed during this study are included in the manuscript and supporting files; Source data files have been provided for Figure 1-7.

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
