## [Editor Report · eLife Assessment]

This study presents a **valuable** finding on the molecular mechanisms that govern GABAergic inhibitory synapse function. The authors propose that Endophilin A1 serves as a novel regulator of GABAergic synapses by acting as a component of the inhibitory postsynaptic density. The findings are **convincing** and likely to interest a broad audience of scientists focusing on inhibitory synaptic transmission, the excitation-inhibition balance, and its disruption in disorders such as epilepsy.

---

## [Referee Report · Reviewer #1 (Public review)]

Summary:

In the present study, Chen et al. investigate the role of Endophilin A1 in regulating GABAergic synapse formation and function. To this end, the authors use constitutive or conditional knockout of Endophilin A1 (EEN1) to assess the consequences on GABAergic synapse composition and function, as well as the outcome for PTZ-induced seizure susceptibility. The authors show that EEN1 KO mice show a higher susceptibility to PTZ-induced seizures, accompanied by a reduction in the GABAergic synaptic scaffolding protein gephyrin as well as specific GABAAR subunits and eIPSCs. The authors then investigate the underlying mechanisms, demonstrating that Endophilin A1 binds directly to gephyrin and GABAAR subunits, and identifying the subdomains of Endophilin A1 that contribute to this effect. Overall, the authors state that their study places Endophilin A1 as a new regulator of GABAergic synapse function.

Strengths:

Overall, the topic of this manuscript is very timely, since there has been substantial recent interest in describing the mechanisms governing inhibitory synaptic transmission at GABAergic synapses. The study will therefore be of interest to a wide audience of neuroscientists studying synaptic transmission and its role in disease. The manuscript is well written and contains a substantial quantity of data. In the revised version of the manuscript, the authors have increased the number of samples analyzed and have significantly improved the statistical analysis, thereby substantially strengthening the conclusions of their study.

---

## [Referee Report · Reviewer #2 (Public review)]

Summary:

The function of neural circuits relies heavily on the balance of excitatory and inhibitory inputs. Particularly, inhibitory inputs are understudied when compared to their excitatory counterparts due to the diversity of inhibitory neurons, their synaptic molecular heterogeneity, and their elusive signature. Thus, insights into these aspects of inhibitory inputs can inform us largely on the functions of neural circuits and the brain.

Endophilin A1, an endocytic protein heavily expressed in neurons, has been implicated in numerous pre- and postsynaptic functions, however largely at excitatory synapses. Thus, whether this crucial protein plays any role in inhibitory synapse, and whether this regulates functions at the synaptic, circuit, or brain level remains to be determined.

The three remaining concerns are:

(1) The use of one-way ANOVA is not well justified.

(2) The use of superplots to show culture to culture variability would make it more transparent.

(3) Change EEN1 in Figure 8B to EndoA1.

Comments on revised version:

The authors addressed the concerns adequately.

---

## [Referee Report · Reviewer #3 (Public review)]

Chen et al. identify endophilin A1 as a novel component of the inhibitory postsynaptic scaffold. Their data show impaired evoked inhibitory synaptic transmission in CA1 neurons of mice lacking endophilin A1, and an increased susceptibility to seizures. Endophilin can interact with the postsynaptic scaffold protein gephyrin and promotes assembly of the inhibitory postsynaptic element. Endophilin A1 is known to play a role in presynaptic terminals and in dendritic spines, but a role for endophilin A1 at inhibitory postsynaptic densities has not yet been described, providing a valuable addition to the field.

To investigate the role of endophilin A1 at inhibitory postsynapses, the authors used a broad array of experimental approaches, including tests of seizure susceptibility, electrophysiology, biochemistry, neuronal culture and image analysis. The authors have addressed the remaining concerns in their revision. Taken together, their results expand the synaptic role of endophilin-A1 to include the inhibitory post synaptic element.

---

## [Author Response]

The following is the authors’ response to the previous reviews

**Reviewer #2 (Recommendations for the authors):**
Comments on revised version:The authors addressed the concerns adequately. The three remaining concerns are:(1) The use of one-way ANOVA is not well justified.

The statement about statistical test in “Statistical analysis” section is as follows in the revised manuscript, “Data sets were tested for normality and direct comparisons between two groups were made using two-tailed Student’s t test (t test, for normally distributed data) as indicated. To evaluate statistical significance of three or more groups of samples, one-way ANOVA analysis with a Tukey test was used or repeated measures ANOVA analysis with a Tukey test was used in behavior assays. Statistical parameters are reported in the figures and the corresponding legends”.

We used a one-way ANOVA for the data about one categorical independent variable and one quantitative dependent variable. The independent variable should have at least three different groups or categories. And we conducted repeated measures ANOVA analysis for the data about behavioral tests according to the suggestion by Reviewer #1 (Point 18) in revised manuscript.

(2) The use of superplots to show culture to culture variability would make it more transparent.

Thanks for the nice suggestion. While superplots could more transparently show culture to culture variability, it is difficult to add more colors or even shades to the scatterplots in the current form, which have already been color coded for multiple groups of samples. The scatterplots we used effectively illustrate the variability across all collected data and do not affect the conclusions of our study. Therefore, we prefer not to change the way of data presentation in the revised manuscript.

(3) Change EEN1 in Figure 8B to EndoA1.

Thanks a lot for the sharp eye. Corrected.

**Reviewer #3 (Recommendations for the authors):**
Specific comments:The authors have made a substantial effort to improve their manuscript. A number of issues, related to numbers of observations mentioned by the reviewers, are clarified in the revised manuscript. The authors have also clarified some of the other questions from the reviewers. The long list of issues brought up by the reviewers and the many corrections needed still raise questions about data quality in this manuscript.In response to my comments (Point 2), the added experiment with PSD95.FingR and GPN.FingR in cultured neurons (Fig. S5A-D) is a good addition; the in vivo data using FingRs in Figure S3 look less convincing however. In response to my Point 5, the authors have added a cell-free binding assay (Figure 5I). This is a useful addition, but to convincingly make the point of interaction between Gephyrin and EndoA1, more rigorous biophysical quantitation of binding is needed. The legend in Figure 5I states that 4 independent experiments were performed, but the graph only shows 3 dots. This needs to be corrected.

We sincerely appreciate your comments and apologize for any concerns raised. As suggested (Point 2), we made many efforts to visualize endogenous postsynaptic proteins using recombinant probes. However, due to much lower expression of GPN.FingR compared with PSD95.FingR in P21 brain slices following viral infection (Figure S3), we were unable to obtain better imaging results. To strengthen our data and conclusions, we additionally performed experiments with PSD95.FingR and GPN.FingR in cultured neurons (Fig. S5A-D) in the revised manuscript.

Regarding the biophysical quantification of gephyrin–endophilin A1 binding, we do not have the equipment for this type of experiment (surface plasmon resonance or isothermal titration calorimetry). Instead, we performed a pull-down assay as an alternative to confirm their interaction (Figure 5I). We also apologize for the error in the number of independent experiments stated in the figure legend and have corrected it in the revised manuscript.